# DVT-LLaVA: Vision-Language Model Personalization with Disentangled Visual Tuning

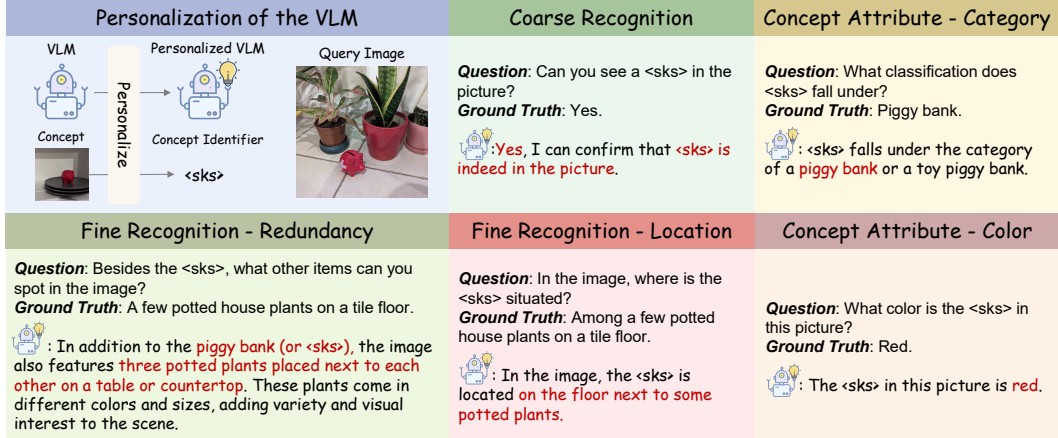

Figure 1: Results of DVT-LLaVA. Given a set of images representing a reference concept, our personalized VLM can perform various visual question-answering (VQA) tasks, including coarse recognition, fine recognition, concept attribute, and personalized caption.

## ABSTRACT

Personalizing foundational vision-language models (VLMs) specifically for individual users could enhance user experience when interacting with VLMs. Most existing methods rely on introducing additional trainable tokens and finetuning VLMs to fit the data of users. Despite the demonstrated improvements on some Visual Question Answering (VQA) benchmarks, we reveal that the improvements come mostly from the shortcut approach to memorizing the information from the introduced textual training dataset. The capability of visually understanding the user's target concepts – key to the VQA tasks – however remains mostly not improved after finetuning. This is especially true for visual concepts residing in complex backgrounds, as these methods often learn representations with concept-relevant and concept-irrelevant information intertwined. To tackle these issues, we introduce DVT-LLaVA, which learns disentangled visual representations for target concepts by jointly learning the concept-relevant tokens and concept-irrelevant tokens via a crafted vision-text dataset derived from image captions. We further propose to tune the LayerNorm layers to enhance the learning capacity and adopt a text embedding augmentation strategy to mitigate overfitting on the training text-image pairs. In addition, we reveal that the existing evaluation benchmarks in this field are mainly based on multiple-choice questions, which fail to accurately assess model performance in the open-set setting. To remedy this, we establish a new benchmark to evaluate performance on this aspect. Extensive evaluations demonstrate the superiority and versatility of DVT-LLaVA.

## 1 INTRODUCTION

Recent advancements in foundational vision-language models (VLMs) Li et al. (2023b); Zhang et al. (2023a); Gong et al. (2023); Ye et al. (2023); Zhu et al. (2023); Liu et al. (2023); Li et al. (2023a); Pi et al. (2023); Su et al. (2023); Luo et al. (2023a); Zhang et al. (2023b); Gao et al. (2023) bring improvements in various visual tasks, such as image captioning and visual question answering. Despite their broad capabilities, users often desire to get feedback on specific concepts like beloved

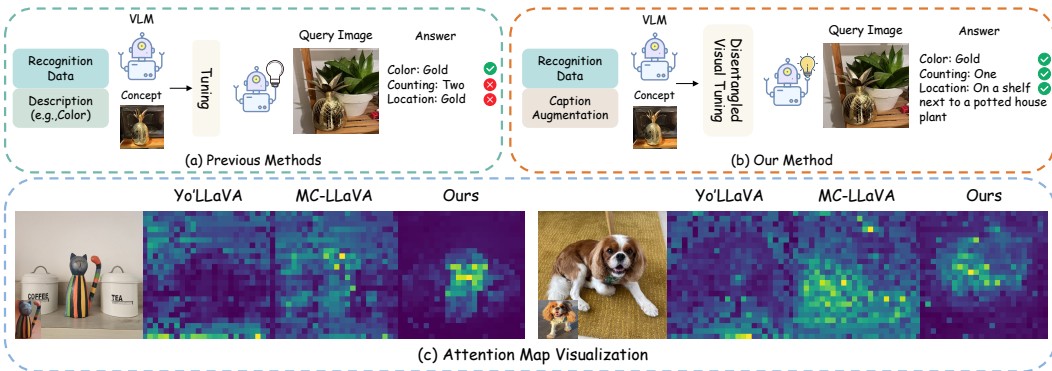

Figure 2: (**a**) and (**b**): Comparison between our method and previous methods. (**c**): Visualization of attention maps of the concept representation learned by different methods.

pets or personal items. However, due to the subjective nature of these personal concepts, they are not captured during the training of the fundamental models, emphasizing the need for the personalization of VLMs. The objective of VLM personalization is to teach the pretrained VLMs to learn a given concept with a few user-provided images and enhance the user experience when interacting with the personalized VLMs.

Early implementations of VLM personalization typically rely on prompting techniques Gu et al. (2023), which are complex and imprecise. By introducing trainable tokens, a series of methods Alaluf et al. (2024); Nguyen et al. (2024); An et al. (2024) have achieved remarkable improvements on some Visual Question Answering (VQA) benchmarks. However, we find that these improvements primarily come from memorizing the information of the textual description in the training data instead of learning the visual feature of the target concept. As shown in Fig. 2 (a), after finetuning on textual description data, the models can easily tell the color of the target concept. In contrast, they fail directly when facing questions requiring visual understanding, such as counting and locating. Moreover, when dealing with concepts in complex environments, these methods struggle to accurately learn the visual features of the target concept and are often disrupted by concept-irrelevant information. As shown in Fig. 2 (c), the representations they obtained failed to establish accurate correspondences with the target concept, thereby further hindering their visual comprehension of the target concept.

To address these issues, we propose DVT-LLAVA, a novel framework of VLM personalization that enhances the visual learning capability of the user's concept with disentangled visual tuning. Our disentangled visual tuning consists of the disentangled visual representation learning, LayerNorm tuning, and the text embedding augmentation. We learn disentangled visual representations by introducing an additional learnable concept-irrelevant token for each training image to capture the concept-irrelevant information of the complex environment, thereby facilitating the preservation of concept-relevant information within the target concept representation. These tokens are learned jointly via a vision-text dataset derived from image captions, without including any textual description of the target concept. We further finetune the LayerNorm layers to enhance the model's visual learning capacity of complex target concepts. Finally, we propose a text embedding augmentation strategy, which introduces noise to the language embedding tokens of queries to mitigate overfitting on the training text-image pairs during personalization.

Since the personalization of VLMs is a relatively new task, most previous methods employ multiple-choice questions for evaluation, which fail to accurately assess model performance in the open-set setting. To address this issue, we propose the OPBench, which constructs all evaluation questions in an open-style VQA format without any options. We also design the judgment prompt to evaluate the accuracy and quality of each answer in a zero-shot manner.

Through extensive qualitative and quantitative comparisons, we demonstrate the effectiveness and superiority of our method. We also conduct comprehensive ablation studies to verify the effectiveness of each component of our DVT-LLAVA. Our contributions are summarized as follows:

- We propose DVT-LLAVA, a novel framework of VLM personalization that enhances the model's visual understanding of the user's concept with disentangled visual tuning.
- DVT-LLAVA learns disentangled visual representations of the user's concept by jointly learning the concept-relevant and concept-irrelevant tokens with a crafted dataset. We

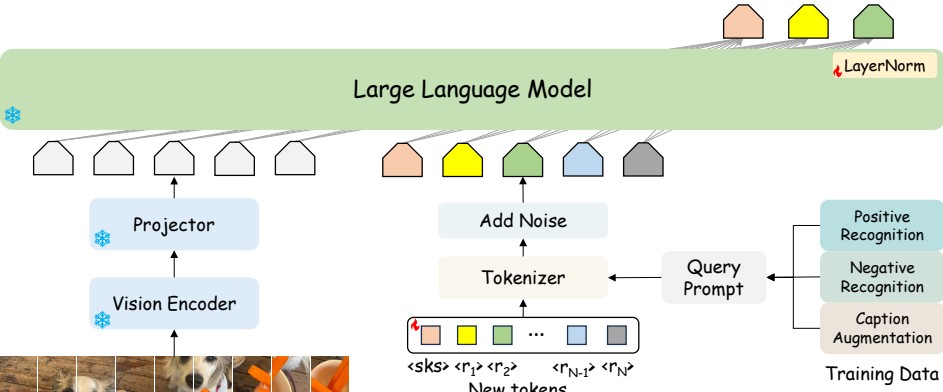

Figure 3: Overall pipeline of DVT-LLAVA

further tune the LayerNorm layers to enhance learning capacity and adopt a text embedding augmentation strategy to mitigate overfitting.
- To provide a comprehensive evaluation for VLM personalization, we introduce the OPBench, an open-style benchmark for VLM personalization. Extensive qualitative and quantitative evaluations demonstrate the effectiveness and superiority of our DVT-LLAVA.

## 2 RELATED WORK

### 2.1 LARGE MULTIMODAL MODELS

Vision-language models (VLMs), represented by GPT-4V Achiam et al. (2023), have become a prominent research focus. These models leverage powerful Large Language Models (LLMs) Chiang et al. (2023); Touvron et al. (2023); Zhao et al. (2023) as their central processing units to accomplish multimodal tasks. Based on their approaches to leveraging LLMs, we categorize them into three types: query-based, projection-based, and PEFT-based methods. Query-based methods Li et al. (2023b); Zhang et al. (2023a); Gong et al. (2023); Ye et al. (2023); Zhu et al. (2023) utilize a set of learnable query tokens to interact with the image, thereby facilitating the fusion of information between text and image. Projection-based methods Liu et al. (2023); Li et al. (2023a); Pi et al. (2023); Su et al. (2023) typically use a projection module to map visual information into the semantic space. PEFT-based methods Luo et al. (2023a); Zhang et al. (2023b); Gao et al. (2023) achieve visual capabilities by fine-tuning specific parameters or incorporating external adapters. In this paper, we mainly focus on personalizing these foundational VLMs for individual users, which could enhance their experience when interacting with VLMs.

### 2.2 PERSONALIZATION OF VLM

The task of VLM personalization is first introduced by MyVLM Alaluf et al. (2024), which employs an additional visual head to detect the presence of personalized concepts and determine whether to load personalized embeddings. Yo'LLaVA Nguyen et al. (2024) extends the token vocabulary by incorporating extra concept tokens and soft tokens, utilizing diverse training data to personalize the VLM. MC-LLaVA An et al. (2024) focuses on multi-concept personalization in VLMs and explores concept token initialization. CaT An et al. (2025b) improves previous tuning-based methods with synthetic data. RAP Hao et al. (2024) and PeKit Seifi et al. (2025) implement personalized multimodal tasks using a retrieval-augmented generation (RAG) approach. Meanwhile, PViT Pi et al. (2024) and PLVM Pham et al. (2024) supported reference-based personalization of VLMs through fine-tuning on large-scale datasets. We improve the existing methods in the following aspects. Firstly, we learn both the concept-relevant and concept-irrelevant tokens to obtain the disentangled visual representation of the target concept. Secondly, we tune the LayerNorm layers instead of using soft tokens to enhance the learning capacity. Thirdly, we introduce noise to the language embedding tokens of queries during training, thereby preventing the model from repeatedly overfitting the same question-answer pair.

Figure 4: When training with textual descriptions, previous methods can easily answer the description questions but struggle with simple visual questions like counting. When training without textual descriptions, these methods directly fail in description questions.

## 3 METHOD

Given a set of images $\{I^i\}_{i=1}^N$ of a target concept (e.g., five images of an user's dog), the objective of vision-language models (VLMs) personalization is to finetune a pre-trained VLM (e.g., LLaVA Liu et al. (2023)) with the images so that the user can interact with the VLM about the concept. We propose DVT-LLAVA which jointly learns concept-relevant and concept-irrelevant tokens to obtain the disentangled visual representation of the target concept. The training data for these newly added tokens is collected through caption augmentation, without textual descriptions of the target concept. We further propose the LayerNorm tuning and text embedding augmentation strategies to enhance the learning capacity while mitigating overfitting during training. An overview of our method is presented in Fig. 3.

### 3.1 SHORTCUT PATH TO IMPROVEMENT BY MEMORIZING TEXTURAL INFORMATION

Previous methods Alaluf et al. (2024); Nguyen et al. (2024); An et al. (2024) mainly use a newly added language embedding token (e.g., <sks>) as the representation of the target concept. They collect the following three types of data to finetune and personalize the VLM: (1) *Positive Recognition*, which teaches the model to identify the presence of target concept in an image; (2) *Negative Recognition*, which helps the model determine when target concept is absent; and (3) *Textual Descriptions*, which provide description-related information about target concept (e.g., the color or category of the target concept), typically generated using LLaVA Liu et al. (2023) or external models like GPT-4V Achiam et al. (2023). When training with the above data, we find that previous methods tend to take a shortcut path to improvement by memorizing the information of textual descriptions. As shown in Fig. 4, previous methods rely on textual descriptions to answer description questions (second and fourth columns) but struggle with simple visual questions such as counting (third column). Moreover, this shortcut learning tendency may hinder the model's visual comprehension of the target concept, particularly when the concept is situated in complex environments containing concept-irrelevant objects. In this case, both the concept-relevant and concept-irrelevant information can be entangled into the representation of the target concepts.

### 3.2 DISENTANGLED VISUAL REPRESENTATION LEARNING

To circumvent the shortcut effect, we propose to learn disentangled visual representations for the concept. We jointly learn both the concept-relevant tokens and the concept-irrelevant tokens to preserve the concept-relevant information within the representation of the target concept. These tokens are learned with a vision-text dataset derived from image captions, without including any textual descriptions of the target concept.

For each image $\{I^i\}_{i=1}^N$ in the training set, where $N$ is the number of training images, we assign an additional text tokens $\{<r_i>\}_{i=1}^N$ as its concept-irrelevant tokens. For all the images, they share a single concept-relevant token <sks>. We jointly train the concept-irrelevant tokens $\{<r_i>\}_{i=1}^N$ and concept-relevant tokens <sks>. Finally, the concept-relevant tokens $s_k$ form our disentangled visual representation of the target concept. The training process employs a vision-text dataset containing both token types but excludes any textual descriptions of the target concept. We construct this dataset in the form of visual question answering using the captions of the training images. As shown in Fig. 5, we first obtain the caption of the training image $I^i$ with the VLM. The object that appears repeatedly in all images is used as the target concept. Then we rewrite and augment the caption to obtain the following three types of data: (1) Localization of the target concept. We guide the model to localize the target concept by constructing location-based question-answering pairs. (2) Description of redundant objects. Since each image contains unique redundant objects, we employ direct visual descriptions to enable the model to develop a thorough understanding of them. (3) Interaction between

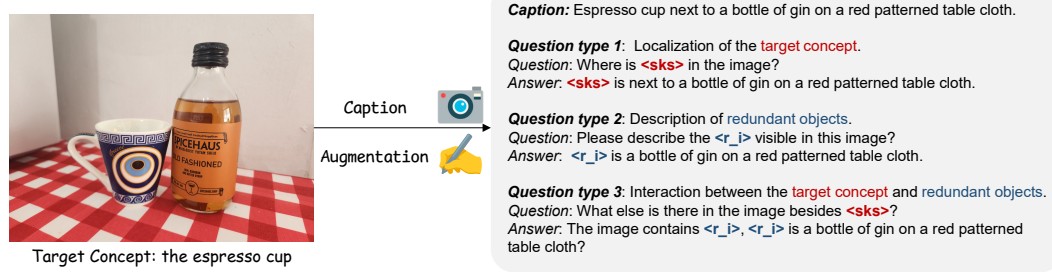

Figure 5: Overview of collecting training data through caption augmentation.

the target concept and redundant objects. Since the model has developed a thorough understanding of the redundant objects, we further promote the learning of the target concept through their interaction. In addition to the above data, we also include the positive recognition and negative recognition data mentioned in Sec. 3.1 to improve the model's basic recognition ability of the target concept.

Our method accurately transfers the visual information of the target concept in images into disentangled tokens, enhancing the model's understanding of the target concept while preventing shortcut learning by memorizing textual information. Additionally, we also quantitatively and qualitatively evaluate the effectiveness of this method in Sec. 4.4.

### 3.3 LAYERNORM TUNING

After collecting the data for personalization, we turn our attention to the model parameters for personalization. Previous VLM personalization methods Alaluf et al. (2024); Nguyen et al. (2024); An et al. (2024) only finetune several newly added tokens. These tokens are insufficient for learning a complex target concept, which is demonstrated by many image personalization methods Kumari et al. (2023); Ruiz et al. (2023); Wei et al. (2023). To further enhance the learning capacity of our method, we propose the LayerNorm tuning strategy, which finetunes the LayerNorm Ba et al. (2016) layers within the VLM. Following previous methods Alaluf et al. (2024); Nguyen et al. (2024); An et al. (2024), we also finetune the newly added tokens <sks> and $\{<r_i>\}_{i=1}^N$. We expand the head matrix $H$ of the LLM $f_\phi$ from $D \times M$ to $D \times (M + 1)$ for the output of the concept identifier <sks>, where $D$ is the feature dimension and $M$ is the vocabulary size. In our framework, the final tunable parameters are:

$$\boldsymbol{\theta} = \left\{ \texttt{<sks>}, <r_1>, <r_2>, \ldots, <r_N>, H_{(:,M+1)}, \gamma \right\}, \tag{1}$$

where $\gamma$ stands for the LayerNorm parameters. All parameters in the VLM are frozen except for the tunable parameters. We employ the original autoregressive training objective to finetune the VLM:

$$p\left(X_a \mid I, X_q\right) = \prod_{i=1}^{L} p_{\boldsymbol{\theta}}\left(x_i \mid I, X_{q,<i}, X_{a,<i}\right), \tag{2}$$

where $I$ denotes the input image, $X_q$ represents the language instruction, $X_a$ is the target answer, $\boldsymbol{\theta}$ represents the trainable parameters, and $X_{q,<i}$ and $X_{a,<i}$ denote the instruction and answer tokens preceding the current prediction token $x_i$.

Tuning LayerNorm layers is a simple yet effective strategy to enhance the learning capability during personalization. Since LayerNorm accounts for only a small part of the model's parameters, training these parameters incurs minimal computational cost and does not lead to obvious catastrophic forgetting. We also compare the tuning results of LayerNorm layers and other types of parameters of VLM. All the related results can be found in Sec. 4.5

### 3.4 MITIGATE OVERFITTING DURING PERSONALIZATION

Although our DVT effectively enhances the model's visual learning ability, we observe that the model tends to overfit specific question-answer patterns due to limited training data, as shown in Fig. 6. Similar phenomena have also been found in some recent studies Qiu et al. (2024); Yan et al. (2025); Sakarvadia et al.

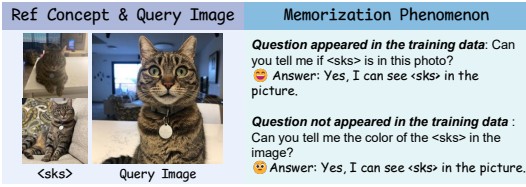

Figure 6: Memorization during personalization.

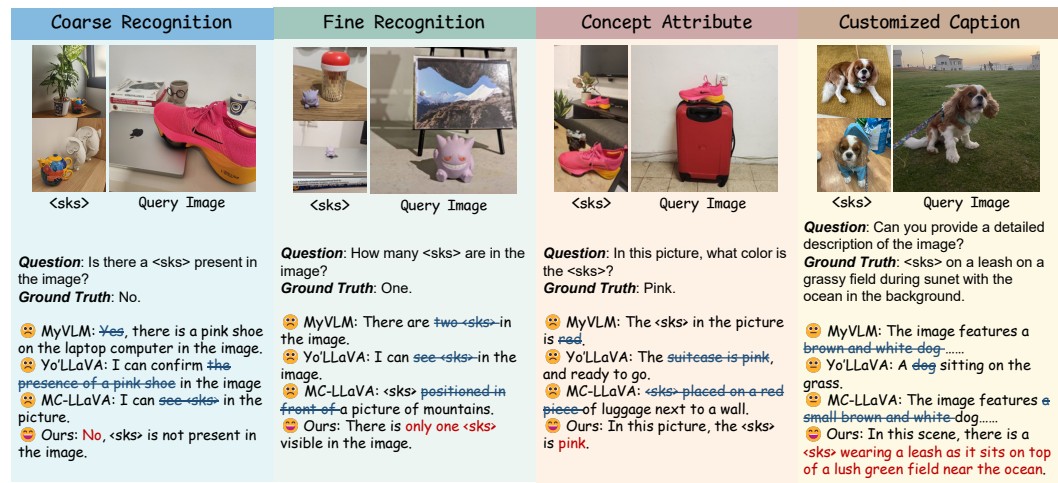

Figure 7: Qualitative comparisons between our DVT-LLAVA and other methods.

(2024). To address this issue, we propose a text embedding augmentation strategy during the training phase. This strategy introduces noise to the language embedding tokens of queries to enhance the diversity of the input embeddings, thus mitigating overfitting during personalization.

Given a standard VQA training triplet $(I^i, X_q^i, X_a^i)$, the model first converts the image $I^i$ and the query $X_q^i$ into their corresponding embedding vectors $e_v^i$ and $e_q^i$. Unlike the standard training procedure, we add noise $\epsilon$ to the query embedding $e_q^i$ as shown in Fig. 3. The noise $\epsilon$ is sampled uniformly and independently from the interval [-1, 1]. Following Kong et al. (2022); Jain et al. (2023); Zhu et al. (2019), we scale the noise $\epsilon$ by a weight of $\alpha/\sqrt{Ld}$ and obtain the final question embedding through:

$$\hat{e}_q^i = e_q^i + \frac{\alpha}{\sqrt{Ld}}\epsilon \tag{3}$$

where $L$ is the sequence length, $d$ is the embedding dimension, and $\alpha$ is the noise weight factor. By adding noise to query embeddings, we generate variations of query inputs during training without incurring additional costs, thereby preventing the model from repeatedly memorizing the same question-answer pair. This text embedding augmentation strategy effectively mitigates overfitting during personalization. We further discuss the impact of the noise weight factor $\alpha$ in Sec. 4.4.

### 3.5 OPBENCH

Previous methods Nguyen et al. (2024); An et al. (2024) mainly employ multiple-choice questions to construct the benchmark for evaluation. However, recent studies Myrzakhan et al. (2024) show that multiple-choice questions fail to accurately assess LLM performance. When evaluating personalized VLMs, we encounter similar issues with multiple-choice questions. Specifically, MyVLM Alaluf et al. (2024) is fundamentally incapable of correctly responding to negative recognition questions, so its accuracy on these questions should approach zero. Yet when evaluated using multiple-choice questions, it achieves about 20% accuracy (see Sec. B), which significantly misrepresents the model's actual capabilities.

To overcome this problem and conduct a fair evaluation, we propose the open-style personalization benchmark (OPBench). We construct all evaluation questions in an open-style VQA format without providing any options. In this way, we can evaluate personalized VLMs' ability to handle real-world visual question answering tasks. In OPBench, we design eight types of questions from four different aspects for evaluation: (1) Coarse Recognition: (a) Positive Recognition; (b) Negative Recognition. (2) Fine Recognition: (a) Counting; (b) Redundant Object Recognition; (c) Location. (3) Concept Attribute: (a) Color; (b) Category. (4) Personalized Caption. For each question, we either use GPT-4o to generate answers or manually annotate them to obtain the ground truth. Inspired by the development of LLM-as-a-judge Zhang et al. (2024); Lin & Chen (2023); Chiang & Lee (2023), we carefully design a judgment prompt (see Sec. D) to motivate the LLM to evaluate the accuracy and quality of each response in a zero-shot manner. We collect a total of 335 images covering 28 different concepts for our method and ensure each image contains both the target concept and concept-irrelevant objects.

Table 1: Quantitative comparisons. The best and second best results are in **red** and **blue**, respectively.

| Method | Coarse Recognition | | Fine Recognition | | Concept Attribute | | Personalized Caption | | Average | |
|---|---|---|---|---|---|---|---|---|---|---|
| | Accuracy | Score | Accuracy | Score | Accuracy | Score | Accuracy | Score | Accuracy | Score |
| MyVLM | 48.0 | 2.38 | 22.1 | 1.74 | **26.3** | **1.65** | 12.8 | 1.91 | **28.5** | **1.90** |
| Yo'LLaVA | **56.1** | **2.90** | 23.2 | 1.74 | 9.1 | 0.75 | **17.9** | 1.97 | 27.2 | 1.81 |
| MC-LLaVA | 55.8 | 2.86 | **24.4** | **1.83** | 4.0 | 0.40 | **16.3** | **2.07** | 26.1 | 1.76 |
| Ours | **65.3** | **3.24** | **48.0** | **2.90** | **30.0** | **1.87** | 12.5 | **1.99** | **43.4** | **2.61** |

Table 2: Ablation study on different components of our method.

| Method | Coarse Recognition | | Fine Recognition | | Concept Attribute | | Personalized Caption | | Average | |
|---|---|---|---|---|---|---|---|---|---|---|
| | Accuracy | Score | Accuracy | Score | Accuracy | Score | Accuracy | Score | Accuracy | Score |
| Yo'LLaVA (Baseline) | 56.1 | 2.90 | 23.2 | 1.74 | 9.1 | 0.75 | 17.9 | 1.97 | 27.2 | 1.81 |
| + LayerNorm Tuning | 62.9 | 3.17 | 42.5 | 2.59 | 17.7 | 1.27 | 22.5 | 2.25 | 38.9 | 2.36 |
| + Text Embedding Augmentation | 62.3 | 3.08 | 46.6 | 2.87 | 29.1 | 1.86 | 13.9 | 2.08 | 42.1 | 2.57 |
| + Concept-irrelevant Tokens | 65.3 | 3.24 | 48.0 | 2.90 | 30.0 | 1.87 | 12.5 | 1.99 | 43.4 | 2.61 |

We construct the training set with 258 images as mentioned in Sec. 3.2. For our OPBench, we use the rest 77 images as positive evaluation examples. For each positive evaluation example, we include seven types of questions mentioned above (excluding negative recognition), with five sub-questions per type. For each concept, we utilize three images of other concepts as negative evaluation examples and formulate negative recognition questions. As a result, our benchmark comprises a total of 3,115 open-style questions for evaluation. More details of our OPBench can be found in Sec. R.

## 4 EXPERIMENTS

### 4.1 IMPLEMENTATION DETAILS

We employ LLaVA-1.5-13B Liu et al. (2023) as our base model, setting $\alpha$ to 25, leraning rate to 0.001, and training for 10 epochs. Additionally, we report results obtained using LLaVA-1.5-7B as an alternative base model in Sec. K. All experiments are conducted on a single A100-80G GPU.

### 4.2 QUALITATIVE COMPARISON

We compare our method with previous personalized VLMs personalization methods, including MyVLM Alaluf et al. (2024), Yo'LLaVA Nguyen et al. (2024), and MC-LLaVA An et al. (2024). We exclude PVIT Pi et al. (2024) and PLVM Pham et al. (2024) because their code is not publicly available or incomplete. We also exclude RAP Hao et al. (2024) as it requires additional data to construct a retrieval database. All compared methods and our method are trained on the dataset mentioned in Sec. 3.2. Qualitative results are shown in Fig. 7. Our method can accurately and flexibly respond to various user queries using the concept identifier <sks>, even when redundant objects are present in the image. More qualitative results can be found in Sec. M.

### 4.3 QUANTITATIVE COMPARISON

We also conduct a comprehensive quantitative comparison on OPBench. As mentioned in Sec. 3.5, we evaluate all the methods from four aspects: Coarse Recognition, Fine Recognition, Concept Attribute, and Personalized Caption. For each evaluation triplet $\left(I^i, X_q^i, X_a^i\right)$ sampled from OPBench, where $I^i$ and $X_q^i$ are the input image and the query text, $X_a^i$ is the ground truth. We obtain the model's response $\hat{X}_a^i$ with $I^i$ and $X_q^i$. We then use $(X_q^i, X_a^i, \hat{X}_a^i)$ along with our carefully designed judgment prompt (see Sec. D) to query GPT-4o-mini Hurst et al. (2024), obtaining the accuracy (e.g., yes or no) and score (e.g., 0-5) for each response. Finally, we calculate the average accuracy and score across all questions. Quantitative results are shown in Tab. 1. Since the dataset does not directly contain training data related to captions, all methods exhibit limited capabilities in generating personalized captions. Our method demonstrates significant advantages in coarse recognition, fine recognition, and concept attribute, indicating its strong capability in capturing the visual features of target concepts. We also provide additional quantitative results including evaluation with open-source LLM (see Sec. M), human studies (see Sec. O), and evaluation results on other benchmarks (see Sec. C).

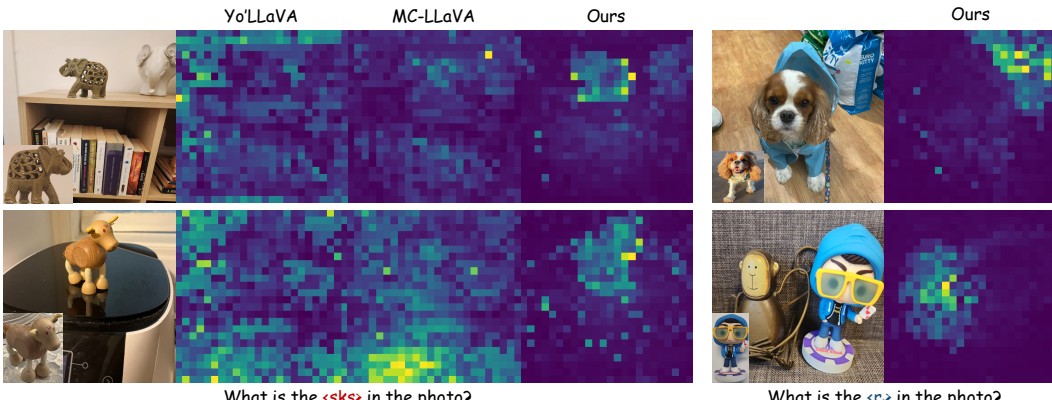

Figure 8: Visualization of attention maps of the concept-relevant token `<sks>` and concept-irrelevant tokens $<r_i>$. The reference target concept is shown in the lower left corner.

Table 3: Catastrophic Forgetting Evaluation.

| Methods | POPE | | | | | RealWorld | ChartQA | | |
| --- | --- | --- | --- | --- | --- | --- | --- | --- | --- |
| | Accuracy | Score | Precision | Recall | Yes Ratio | Accuracy | Aug | Human | Overall |
| Vanilla LLaVA | 0.86 | 0.84 | 0.95 | 0.76 | 0.50 | 0.56 | 0.16 | 0.22 | 0.19 |
| Yo'LLaVA | 0.86 | 0.84 | 0.95 | 0.76 | 0.50 | 0.56 | 0.16 | 0.21 | 0.19 |
| MC-LLaVA | 0.86 | 0.84 | 0.94 | 0.75 | 0.50 | 0.56 | 0.16 | 0.21 | 0.19 |
| Ours | 0.85 | 0.83 | 0.93 | 0.75 | 0.50 | 0.55 | 0.16 | 0.20 | 0.18 |

## 4.4 ABLATION STUDY

**LayerNorm tuning**. We verify the effectiveness of our LayerNorm Tuning strategy. As shown in the second row of Tab. 2, we employ LayerNorm Tuning to replace the soft tokens used in the baseline method. Compared to the baseline approach, LayerNorm Tuning demonstrates significant improvements across all evaluation metrics.

**Text embedding augmentation**. We evaluate the effectiveness of our proposed text embedding augmentation strategy. As shown in the third row of Tab. 2, the introduction of this strategy significantly improves both the Fine Recognition and Concept Attribute metrics. We further conduct a detailed ablation study with different noise weight factors $\alpha$ and visualize the results in Fig. 9. The $x$-axis represents varying $\alpha$ values, while the $y$-axis displays two evaluation metrics: accuracy and score. We observe the following: (1) Incorporating text embedding augmentation always yields superior performance compared to models without text embedding augmentation ($\alpha = 0$). (2) The advantage of text embedding augmentation initially increases and then decreases with the growth of $\alpha$, reaching its peak when $\alpha = 25$. Based on these findings, we select $\alpha = 25$ for our final model.

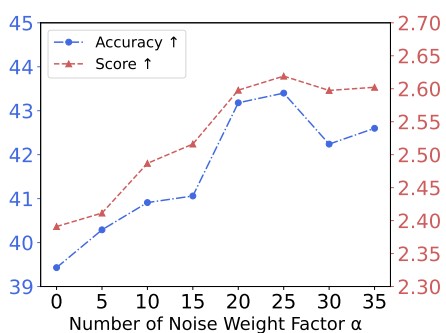

Figure 9: Quantitative results on the ablation study of the noisy weight factor $\alpha$.

**Disentangled visual representation learning**. To verify the effectiveness of our DVRL, we conduct an ablation study by removing the concept-irrelevant tokens from our method. As shown in the fourth row of Tab. 2, the absence of the concept-irrelevant tokens leads to a consistent decline in performance across all evaluation metrics. We also visualize the self-attention map of the disentangled representation `<sks>` and concept-irrelevant token $<r_i>$ in the VLM following Zhang et al. (2025). As shown in Fig. 8, `<sks>` attends to the region of the target concept while $<r_i>$ attends to the region of the redundant objects.

## 4.5 FURTHER DISCUSSION

**Catastrophic forgetting**. Catastrophic forgetting Luo et al. (2023b); Qi et al. (2023); Kirkpatrick et al. (2017) refers to the phenomenon where a neural network (e.g., VLMs), after being trained on a

Table 4: Quantitative results on multiple choice benchmark.

| Method | Coarse Recognition | | Fine Recognition | | Concept Attribute | | personalized Caption | | Average | |
|---|---|---|---|---|---|---|---|---|---|---|
| | Accuracy | Score | Accuracy | Score | Accuracy | Score | Accuracy | Score | Accuracy | Score |
| Yo'LLaVA | 34.1 | 2.01 | 44.4 | 2.43 | 21.6 | 1.18 | 45.0 | 2.33 | 36.2 | 2.00 |
| MC-LLaVA | 42.5 | 2.12 | 61.6 | 3.44 | 32.5 | 1.62 | 63.3 | 3.21 | 49.7 | 2.63 |
| Ours | 39.1 | 2.34 | 82.7 | 4.27 | 60.0 | 3.02 | 66.6 | 3.65 | 64.1 | 3.40 |

Table 5: Quantitative comparisons when using textual datasets.

| Method | Dataset | Coarse Recognition | | Fine Recognition | | Concept Attribute | | personalized Caption | | Average | |
|---|---|---|---|---|---|---|---|---|---|---|---|
| | | Accuracy | Score | Accuracy | Score | Accuracy | Score | Accuracy | Score | Accuracy | Score |
| MyVLM | w/ Description | 46.9 | 2.31 | 26.3 | 1.96 | 37.0 | 2.26 | 12.2 | 1.92 | 30.6 | 2.11 |
| Yo'LLaVA | w/ Description | 67.3 | 3.40 | 12.7 | 1.19 | 15.1 | 1.11 | 14.4 | 2.12 | 27.2 | 1.84 |
| MC-LLaVA | w/ Description | 48.2 | 2.35 | 29.9 | 2.10 | 49.5 | 2.72 | 14.9 | 2.12 | 37.5 | 2.32 |
| Ours | w/o Description | 65.3 | 3.24 | 48.0 | 2.90 | 30.0 | 1.87 | 12.5 | 1.99 | 43.4 | 2.61 |

new task, experiences a dramatic decline in performance on its original task. To evaluate the extent of catastrophic forgetting in our method, we select three representative benchmarks for vision-language models: POPE Li et al. (2023c), RealWorldQA xAI (2024), and ChartQA Masry et al. (2022). These benchmarks assess the performance of a VLM from three perspectives: hallucination, commonsense knowledge, and domain-specific expertise. As shown in Tab. 3, our method maintains almost identical performance compared to Vanilla LLaVA Liu et al. (2023), Yo'LLaVA Nguyen et al. (2024), and MC-LLaVA An et al. (2024) across all three benchmarks, demonstrating that our method effectively balances between learning personalized concepts and preserving the model's pretrained knowledge.

**Results on multiple-choice benchmarks** As most previous methods adopted multiple-choice evaluation, we adjust our OPBench to the multiple-choice format and conduct quantitative experiments. The results are presented in Tab. 4. The results show that our method still outperforms the baseline methods in multiple-choice evaluation.

**Tuning other modules**. We present the results of tuning with different modules in Tab. 6 and Fig. 10. Due to the limitation of the GPU memory, this experiment is conducted with the LLaVA-1.5-7B model. The results demonstrate that when personalizing a VLM, an excessive number of parameters may lead to worse performance.

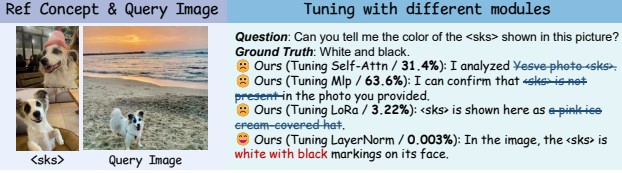

Figure 10: Comparison of the parameter size of different modules and the effect when personalizing with them.

In contrast, LayerNorm requires the fewest parameters and effectively mitigates the problem of overfitting.

**Tuning with the textual dataset as previous methods**. We report the results of training the compared methods using the dataset with textual descriptions. As shown in Tab. 5, all compared methods exhibit a significant improvement in Concept Attribute with textual description data. However, their performance in Fine Recognition remains suboptimal. Notably, even without textual descriptions, our method surpasses the compared methods in terms of both average accuracy and score.

Table 6: Quantitative results of tuning with different modules.

| Method | Coarse Recognition | | Fine Recognition | | Concept Attribute | | personalized Caption | | Average | |
|---|---|---|---|---|---|---|---|---|---|---|
| | Accuracy | Score | Accuracy | Score | Accuracy | Score | Accuracy | Score | Accuracy | Score |
| Ours (Self-Attn) | 2.26 | 0.31 | 0.00 | 1.25 | 0.00 | 0.65 | 0.00 | 0.22 | 0.57 | 0.11 |
| Ours (MLP) | 35.2 | 1.89 | 0.00 | 0.07 | 0.00 | 0.03 | 0.36 | 0.31 | 8.85 | 0.55 |
| Ours (LoRa) | 51.5 | 2.67 | 9.54 | 0.91 | 2.32 | 0.20 | 7.26 | 1.04 | 17.9 | 1.19 |
| Ours (LayerNorm) | **67.0** | **3.36** | **45.7** | **2.86** | **27.2** | **1.73** | **13.5** | **1.94** | **42.4** | **2.59** |

## 5 CONCLUSION

This paper introduces DVT-LLAVA, which learns a disentangled visual representation for the target concept through the joint learning of concept-relevant and concept-irrelevant tokens. Furthermore, we propose the LayerNorm tuning and text embedding augmentation strategies to enhance the learning

capacity while mitigating overfitting during training. Finally, we introduce OPBench for evaluation. The impressive performance of DVT-LLAVA demonstrates its effectiveness.

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

# A    DETAILED IMPLEMENTATIONS OF THE COMPARED METHODS

**MyVLM Alaluf et al. (2024).** We employ the LLaVA version of MyVLM, utilizing its official implementation[1] and adhering to the recommended configurations. The concept head is trained with 500 steps, a batch size of 16, and a learning rate of 0.001. The threshold for the concept head was set to 0.5. We train LLaVA using 100 optimization steps with a learning rate of 1.0, while setting the weight factor to 0.25.

**Yo'LLaVA Nguyen et al. (2024).** We employ the official implementation[2] of Yo'LLaVA and optimize its training process using the Transformers Wolf et al. (2020). Following the recommended configurations, we set the number of soft tokens to 16, the learning rate to 0.001, and trained the model for 10 epochs.

**MC-LLaVA An et al. (2024).** We employ the official implementation[3] of MC-LLaVA and optimize its training process using the Transformers Wolf et al. (2020). Following the recommended configurations, we set the number of concept tokens to 16, the learning rate to 0.001, and trained the model for 15 epochs.

Table 7: Quantitative evaluation of MyVLM Alaluf et al. (2024) on the Negative Recognition questions with different evaluation methods.

| Methods | Number of Correct Answers | Accuracy |
|---|---|---|
| Choise-based Evaluation | 84 | $\sim$20% |
| Our Open-style Evaluation | 6 | $\sim$1.4% |

# B    COMPARISON BETWEEN CHOICE-BASED AND OPEN-STYLE EVALUATION

MyVLM Alaluf et al. (2024) is fundamentally incapable of correctly responding to negative recognition problems because it fails to identify negative examples. Consequently, its accuracy on such tasks should theoretically approach zero. However, as shown in Tab. 7, when evaluated using multiple-choice questions, the model achieves approximately 20% accuracy, which significantly misrepresents its true capabilities. In contrast, our open-style evaluation accurately assesses model performance.

| Evaluation Dataset | MC-LLaVA | | | | | | | | | | | | | | | | Yo'LLaVA |
|---|---|---|---|---|---|---|---|---|---|---|---|---|---|---|---|---|---|
| | Choice-V Acc. | | | Choice-T Acc. | | | VQA BLEU | | | Captioning Recall | | | Rec | | | VG | Average Acc. |
| **Method** | Single | Multi | Weight | Single | Multi | Weight | Single | Multi | Weight | Single | Multi | Weight | Single | Multi | Weight | | Single |
| **MyVLM\*** | 0.779 | - | 0.779 | - | - | - | 0.640 | - | **0.640** | **0.714** | - | **0.714** | 0.795 | - | 0.795 | 0.688 | 77.92 |
| **Yo'LLaVA** | 0.687 | 0.634 | 0.661 | 0.604 | **0.605** | 0.604 | 0.623 | **0.538** | 0.575 | 0.498 | 0.663 | 0.548 | **0.940** | 0.733 | 0.820 | 0.639 | 92.22 |
| **RAP-MLLM\*** | 0.832 | 0.690 | 0.784 | **0.709** | **0.656** | **0.685** | 0.424 | 0.423 | 0.424 | **0.711** | **0.748** | **0.723** | 0.747 | 0.688 | 0.713 | **0.719** | 94.29 |
| **MC-LLaVA** | **0.855** | **0.765** | **0.796** | **0.623** | 0.585 | **0.612** | **0.646** | 0.526 | 0.567 | 0.531 | 0.708 | 0.584 | 0.810 | **0.882** | **0.836** | **0.714** | **95.70** |
| **Ours** | **0.857** | **0.806** | **0.823** | 0.546 | 0.515 | 0.536 | **0.706** | **0.603** | **0.638** | 0.654 | **0.844** | 0.711 | **0.927** | **0.779** | **0.873** | 0.693 | **96.80** |
| **Yo'Chameleon** | 0.718 | - | 0.718 | 0.561 | - | 0.561 | 0.603 | - | 0.603 | 0.602 | - | 0.602 | 0.723 | - | 0.723 | 0.621 | 74.35 |
| **UniCTokens** | 0.753 | - | 0.753 | 0.572 | - | 0.572 | 0.641 | - | 0.641 | 0.634 | - | 0.634 | 0.742 | - | 0.742 | 0.632 | 75.68 |

Table 8: Comparison of our method and other baselines in MC-LLaVA and Yo'LLaVA datasets. **Red** stands for the best result, **Blue** stands for the second best result. \* indicates that the results of this method on MC-LLaVA dataset are referenced from (An et al., 2024).

# C    EVALUATION RESULTS ON OTHER RELATED BENCHMARKS

To achieve a more comprehensive evaluation, we conduct an additional experiment on the public benchmark provided by Yo'LLaVA (Nguyen et al., 2024) and MC-LLaVA (An et al., 2024). These benchmarks are different from our OPBench. Most images in these benchmarks contain only the target concept without the complex backgrounds or redundant objects. Besides, most evaluation is

---

[1] https://github.com/snap-research/MyVLM

[2] https://github.com/WisconsinAIVision/YoLLaVA

[3] https://github.com/arctanxarc/MC-LLaVA

> You are an intelligent chatbot designed for evaluating the correctness of generative outputs for question-answer pairs.
> Your task is to compare the predicted answer with the correct answer and determine if they match meaningfully. Here's how you can accomplish the task:
> ------
> ##INSTRUCTIONS:
> - Focus on the meaningful match between the predicted answer and the correct answer.
> - Consider synonyms or paraphrases as valid matches.
> - Evaluate the correctness of the prediction compared to the answer.
>
> Please evaluate the following video-based question-answer pair:
> Question: {question}
> Correct Answer: {answer}
> Predicted Answer: {predict}
> Provide your evaluation only as a yes/no and score where the score is an integer value between 0 and 5, with 5 indicating the highest meaningful match.
> Please generate the response in the form of a Python dictionary string with keys 'pred' and 'score', where value of 'pred' is  a string of 'yes' or 'no' and value of 'score' is in INTEGER, not STRING.
> DO NOT PROVIDE ANY OTHER OUTPUT TEXT OR EXPLANATION. Only provide the Python dictionary string.
> For example, your response should look like this: {'pred': 'yes', 'score': 4.8}.

Figure 11: Visualization of the judgment prompts.

Table 9: Quantitative evaluation with different model sizes.

| Method | Coarse Recognition | | Fine Recognition | | Concept Attribute | | Personalized Caption | | Average | |
| --- | --- | --- | --- | --- | --- | --- | --- | --- | --- | --- |
| | Accuracy | Score | Accuracy | Score | Accuracy | Score | Accuracy | Score | Accuracy | Score |
| Ours w/ LLaVA-1.5-7B | 67.0 | 3.36 | 45.7 | 2.86 | 27.2 | 1.73 | 13.5 | 1.94 | 42.4 | 2.59 |
| Ours w/ LLaVA-1.5-13B | 65.3 | 3.24 | 48.0 | 2.90 | 30.0 | 1.87 | 12.5 | 1.99 | 43.4 | 2.61 |

based on multiple-choice questions, contrasting with the open-ended VQA format of our OPBench. We include MyVLM(Alaluf et al., 2024), Yo'LLaVA(Nguyen et al., 2024), RAP-MLLM(Hao et al., 2025), MC-LLaVA(An et al., 2024), Yo'Chameleon(Nguyen et al., 2025), and UnicTokens(An et al., 2025a) as the compared baseline methods. Notably, both Yo'Chameleon and UnicTokens are categorized as personalized unified models, which is different from our personalized VLM task. We include these methods for more comprehensive evaluation. We follow the evaluation protocol provided in the official repository[4] of MC-LLaVA and reproduce the evaluation results. To extend our method to multi-concept scenarios, we first follow MC-LLaVA (An et al., 2024) to assign a specific, concept-relevant token to each personalized concept. We then adapt the data format of the MC-LLaVA dataset to align with our training process. The results are shown in Tab. 8. For MC-LLaVA dataset, although not explicitly designed for multi-concept scenarios, our method demonstrates strong performance across various visual evaluation tasks, including Choice-V, VQA, and Rec. Our method also achieves superior performance in the Yo'LLaVA dataset. These results indicate our method's capability to navigate more complex, multi-concept settings.

## D  JUDGMENT PROMPT

We visualize our judgment prompt in Fig. 11.

---

[4]https://github.com/arctanxarc/MC-LLaVA

Table 10: Quantitative analysis of disentanglement.

| Metric | Cosine Similarity | | Feature Mutual Information | | Distance Correlation | |
|---|---|---|---|---|---|---|
| | Mean | Std | Mean | Std | Mean | Std |
| Value | 0.033 | 0.033 | 0.007 | 0.001 | 0.050 | 0.021 |

# E  QUANTITATIVE ANALYSIS OF DISENTANGLEMENT

To gain a deeper understanding of the disentanglement effect between concept-relevant tokens and concept-irrelevant tokens, we quantitatively analyze the embeddings of the trained concept-relevant and concept-irrelevant tokens. We select three metrics to analyze the relationship between concept-relevant and concept-irrelevant tokens: Cosine Similarity, Feature Mutual Information, and Distance Correlation(Székely et al., 2007).

- Cosine Similarity quantifies the similarity between two vectors based on their direction. Its value ranges from -1 to 1, where 1 signifies that the vectors point in the same direction, -1 indicates they are diametrically opposed, and 0 denotes orthogonality. Consequently, when the cosine similarity between two embedding vectors approaches 0, it implies that they are orthogonal within their high-dimensional vector space.
- Feature Mutual Information measures the statistical dependency between two representations, capturing both linear and non-linear relationships. The value of Feature Mutual Information ranges from zero to infinity $[0, \infty)$, where a value approaching zero indicates greater statistical independence between the two representations.
- Distance Correlation is a non-parametric statistic used to measure the dependence between two random vectors. Its value ranges from 0 to 1, where a value of 0 indicates complete statistical independence. Conversely, a value approaching 1 signifies a stronger dependence between the vectors.

For each concept, we calculate the Cosine Similarity, Feature Mutual Information, and Distance Correlation between the concept-relevant and concept-irrelevant tokens. The results are presented in Tab. 10. The results indicate a low similarity and minimal statistical correlation between concept-relevant and concept-irrelevant tokens. This demonstrates that our disentangled visual representation learning approach has effectively disentangled these two components.

# F  ABLATION STUDY OF DECODING PARAMETERS

During the inference process, we employ deterministic sampling, following previous methods to ensure the reproducibility of our results. To analyze the impact of text generation bias on the model's outputs, we conduct an ablation study on key decoding parameters: temperature, number of beams, and length penalty. During the experiment, each parameter is adjusted individually while the others are held at their default values. We evaluate the output with Qwen-2.5-72B-Instruct (Yang et al., 2024) and with the backbone of Phi3 (Abdin et al., 2024). The results are shown in Tabs. 11 to 13. The results indicate that our model performs consistently across different decoding parameters, suggesting that it has robustly learned the visual features of the target concept and is not significantly influenced by text generation bias.

Table 11: Quantitative ablation results of different length penalty.

| Length Penalty | Coarse Recognition | | Fine Recognition | | Concept Attribute | | Personalized Caption | | Average | |
|---|---|---|---|---|---|---|---|---|---|---|
| | Accuracy | Score | Accuracy | Score | Accuracy | Score | Accuracy | Score | Accuracy | Score |
| 0.8 | 73.3 | 3.53 | 53.7 | 2.75 | 28.6 | 1.60 | 9.16 | 0.99 | 46.8 | 2.44 |
| 1.0 | 72.8 | 3.50 | 53.6 | 2.73 | 29.4 | 1.66 | 7.67 | 0.92 | 46.6 | 2.43 |
| 1.2 | 73.0 | 3.51 | 53.8 | 2.75 | 28.7 | 1.62 | 5.17 | 0.84 | 46.2 | 2.43 |

Table 12: Quantitative ablation results of different number of beams.

| Numbers of Beams | Coarse Recognition | | Fine Recognition | | Concept Attribute | | Personalized Caption | | Average | |
|---|---|---|---|---|---|---|---|---|---|---|
| | Accuracy | Score | Accuracy | Score | Accuracy | Score | Accuracy | Score | Accuracy | Score |
| 1 | 69.1 | 3.31 | 54.7 | 2.76 | 29.8 | 1.61 | 8.21 | 1.05 | 46.2 | 2.39 |
| 3 | 72.7 | 3.50 | 53.5 | 2.75 | 28.6 | 1.63 | 7.26 | 0.92 | 46.3 | 2.43 |
| 5 | 72.6 | 3.50 | 53.6 | 2.72 | 27.4 | 1.58 | 6.72 | 0.87 | 45.9 | 2.40 |

Table 13: Quantitative ablation results of different temperatures.

| Temperature | Coarse Recognition | | Fine Recognition | | Concept Attribute | | Personalized Caption | | Average | |
|---|---|---|---|---|---|---|---|---|---|---|
| | Accuracy | Score | Accuracy | Score | Accuracy | Score | Accuracy | Score | Accuracy | Score |
| 0.2 | 70.4 | 3.38 | 51.2 | 2.65 | 26.7 | 1.48 | 8.63 | 0.89 | 44.6 | 2.32 |
| 0.5 | 70.6 | 3.40 | 50.6 | 2.62 | 23.3 | 1.36 | 7.02 | 0.84 | 43.3 | 2.28 |
| 0.8 | 70.7 | 3.37 | 47.0 | 2.48 | 22.7 | 1.32 | 8.15 | 0.88 | 42.0 | 2.21 |

## G  ABLATION STUDY OF DIFFERENT ARCHITECTURES

In this section, we implement our methods on backbones with different architectures and conduct comprehensive ablation studies to evaluate the specific contributions of LayerNorm tuning and text embedding augmentation.

We select two representative architectures as backbones: Phi3(Abdin et al., 2024) and Qwen(Bai et al., 2023). We first perform an ablation study on the noise weight factor $\alpha$ of our text embedding augmentation. The results are summarized in Tabs. 14 and 16. We evaluate the output with Qwen-2.5-72B-InstructYang et al. (2024). The results show that incorporating text embedding augmentation always yields superior performance compared to models without text embedding augmentation. Furthermore, both architectures achieved optimal performance at $\alpha = 25$, which is consistent with the configuration used in our main experiments. This finding further substantiates the robustness and parameter-insensitivity of our proposed text embedding augmentation strategy.

For LayerNorm tuning, we evaluate three distinct configurations to assess the impact of finetuning LayerNorm layers: (1) No, a baseline in which the LayerNorm layers are not finetuned; (2) Half, where only half of the LayerNorm layers are finetuned; and (3) Full, where all LayerNorm layers are finetuned. We show the results in Tabs. 15 and 17. The results demonstrate that finetuning LayerNorm layers consistently yields superior performance compared to the baseline. Furthermore, the Full configuration, which involved finetuning all LayerNorm layers, achieved the optimal results.

Table 14: Quantitative ablation results of $\alpha$ with Phi3 backbone.

| $\alpha$ | Coarse Recognition | | Fine Recognition | | Concept Attribute | | Personalized Caption | | Average | |
|---|---|---|---|---|---|---|---|---|---|---|
| | Accuracy | Score | Accuracy | Score | Accuracy | Score | Accuracy | Score | Accuracy | Score |
| 0 | 64.5 | 3.04 | 45.4 | 2.42 | 21.6 | 1.30 | 5.23 | 0.58 | 39.2 | 2.07 |
| 5 | 65.0 | 3.10 | 49.5 | 2.56 | 20.8 | 1.19 | 4.34 | 0.63 | 40.5 | 2.11 |
| 15 | 67.4 | 3.22 | 54.0 | 2.75 | 22.4 | 1.31 | 2.32 | 0.56 | 43.0 | 2.23 |
| 25 | **69.1** | **3.31** | **54.7** | **2.76** | **29.8** | **1.61** | **8.21** | **1.05** | **46.2** | **2.39** |
| 35 | 68.3 | 3.24 | 52.5 | 2.65 | 18.7 | 1.14 | 8.03 | 1.08 | 42.4 | 2.22 |

## H  ANALYSIS UNDER SIMILAR CONCEPTS

Our OPBench is designed to address a more challenging, real-world setting in which each image contains complex objects. Among our benchmark, some objects are visually similar to the target concept, potentially sharing attributes such as color or shape. We term these objects "concept neighbors". In this case, concept neighbors can interfere with the learning of the target concept's visual features. To evaluate our model's learning ability in the presence of such concept neighbors, we carefully select a subset from our OPBench: OPBench-Sim. OPBench-Sim contains concepts with

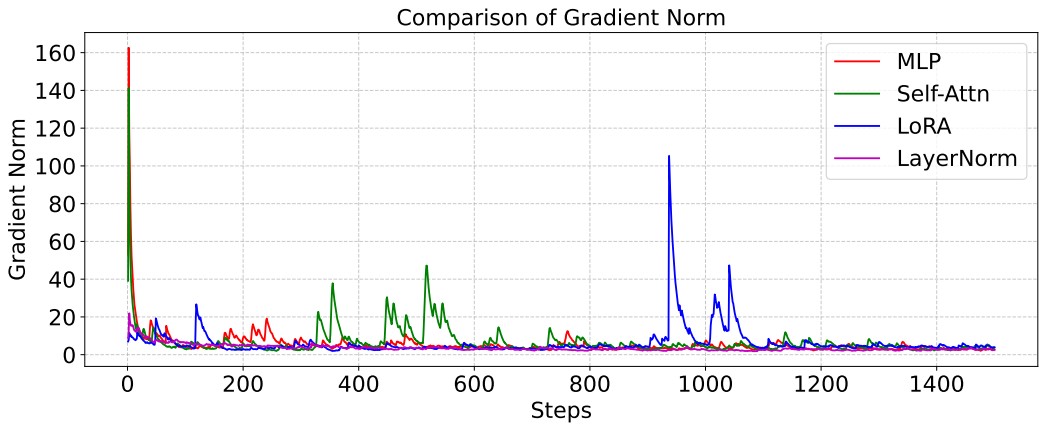

Figure 12: Gradient norm of tuning different parameters.

Table 15: Quantitative ablation results of LayerNorm tuning with Phi3 backbone.

| Layer Norm | Coarse Recognition Accuracy | Score | Fine Recognition Accuracy | Score | Concept Attribute Accuracy | Score | Personalized Caption Accuracy | Score | Average Accuracy | Score |
|---|---|---|---|---|---|---|---|---|---|---|
| No | 43.4 | 2.14 | 14.2 | 0.98 | 4.88 | 0.32 | 5.23 | 0.87 | 18.0 | 1.09 |
| Half | 59.1 | 2.85 | 46.0 | 2.51 | 14.3 | 1.01 | 7.66 | 0.97 | 36.5 | 2.02 |
| Full | **69.1** | **3.31** | **54.7** | **2.76** | **29.8** | **1.61** | **8.21** | **1.05** | **46.2** | **2.39** |

concept neighbors in the evaluation image. With this benchmark, we can gain a deeper understanding of how our method performs when dealing with concept neighbors. The detailed quantitative results are shown in Tab. 18. We also provide a more intuitive result in Fig. 13. Even when confronted with concept neighbors, our method can still accurately learn the visual features of the target concept without significant performance degradation. This outcome highlights our core innovation: learning consistent visual features across diverse scenarios rather than relying on the shortcut of memorizing textual descriptions.

# I    ABLATION STUDY ON THE JUDGMENT PROMPT

We conduct an ablation study on the judgment prompt to assess its impact. We use the context evaluation prompt provided by Video-ChatGPT(Maaz et al., 2024) as the judgment prompt. We use Qwen-2.5-72B-Instruct(Yang et al., 2024) to evaluate the results. The results are presented in Tab. 19. The results demonstrate similar trends to the results we present in Sec. N under all metrics.

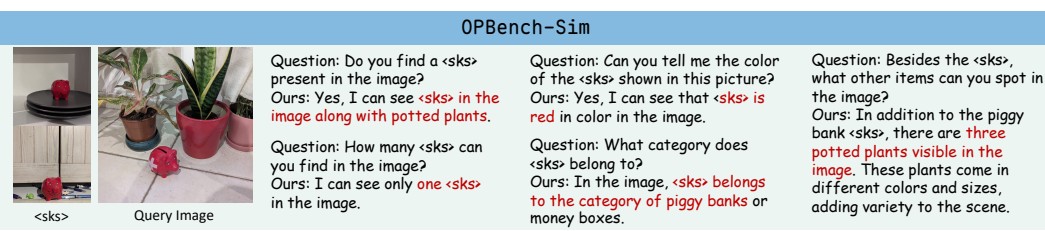

Figure 13: Visualized results on the OPBench-Sim.

Table 16: Quantitative ablation results of $\alpha$ with Qwen backbone.

| $\alpha$ | Coarse Recognition Accuracy | Score | Fine Recognition Accuracy | Score | Concept Attribute Accuracy | Score | Personalized Caption Accuracy | Score | Average Accuracy | Score |
|---|---|---|---|---|---|---|---|---|---|---|
| 0 | 78.1 | 3.83 | 51.8 | 2.62 | 27.3 | 1.50 | **18.6** | **1.54** | 48.1 | 2.50 |
| 5 | 75.7 | 3.71 | 52.3 | 2.65 | 31.1 | 1.63 | 15.8 | 1.51 | 48.3 | 2.52 |
| 15 | 79.8 | 3.77 | 63.3 | 3.03 | 37.4 | 1.97 | 14.5 | 1.50 | 54.9 | 2.76 |
| 25 | **88.3** | **4.11** | **63.4** | **3.12** | 37.7 | **2.08** | 12.2 | 1.33 | **56.8** | **2.88** |
| 35 | 86.8 | 3.95 | 56.1 | 2.79 | **39.2** | 2.00 | 9.82 | 1.32 | 53.7 | 2.70 |

Table 17: Quantitative ablation results of LayerNorm tuning with Qwen backbone.

| Layer Norm | Coarse Recognition Accuracy | Score | Fine Recognition Accuracy | Score | Concept Attribute Accuracy | Score | Personalized Caption Accuracy | Score | Average Accuracy | Score |
|---|---|---|---|---|---|---|---|---|---|---|
| No | 57.1 | 2.80 | 25.3 | 1.53 | 14.0 | 0.84 | 10.0 | 1.25 | 28.5 | 1.64 |
| Half | 79.3 | 3.72 | 49.7 | 2.75 | 28.1 | 1.47 | 17.6 | 1.49 | 47.7 | 2.51 |
| Full | **88.3** | **4.11** | **63.4** | **3.12** | **37.7** | **2.08** | **12.2** | **1.33** | **56.8** | **2.88** |

# J  THEORETICAL ANALYSIS OF LAYERNORM TUNING

To understand the theoretical mechanism of LayerNorm tuning, we investigate the impact of Layer-Norm tuning on gradient dynamics during training in this section. We begin by illustrating the trend of gradients under various training strategies. Subsequently, we present a detailed theoretical analysis of LayerNorm tuning. Our finding demonstrates that LayerNorm tuning performs *recentering and rescaling* to the gradients, thus stabilizing gradients throughout the training process.

During the personalization of VLMs, the gradient serves as a crucial indicator of the training process. Stable gradients(Alman & Song, 2024; Kumar et al., 2025) are essential for ensuring training stability and accelerating model convergence. To analyze this, we monitor the squared norm of the gradient (henceforth referred to as the gradient norm) under various training approaches (e.g., Self-attn, MLP, LoRa, and LayerNorm tuning). This metric is computed by concatenating the gradients of all trainable parameters into a single vector and then calculating its squared L2 norm. The results are visualized in Fig. 12. The results indicate that the gradient norm associated with LayerNorm parameters is smaller and more stable during training compared to that of other parameters. This stability facilitates the efficient learning of visual features for target concepts. Subsequently, we will present a theoretical analysis of the underlying reasons for this advantage in LayerNorm tuning.

For a given input vector $x = (x_1, \ldots, x_H)^T$ and its corresponding output vector $y = (y_1, \ldots, y_H)^T$, the backpropagation process with respect to the loss function $l$ is as follows:

$$\frac{\partial l}{\partial x} = \frac{dy}{dx}\frac{\partial l}{\partial y}. \tag{4}$$

**Theorem 1.** *Let* $b = \frac{\partial l}{\partial x} = (b_1, ..., b_H)^T$. *Its mean is* $\overline{b} = \frac{1}{H}\sum_{i=1}^{H} b_i$, *and its variance is* $D_b = \frac{1}{H}\sum_{i=1}^{H}(b_i - \overline{b})^2$. *The gradient norm* $\|b\|^2$, *can be expressed as:*

$$\|b\|^2 = H \cdot (D_b + \overline{b}^2). \tag{5}$$

Table 18: Quantitative results of our method under OPBench and its subset.

| OPBench | Coarse Recognition | | Fine Recognition | | Concept Attribute | | Personalized Caption | | Average | |
|---|---|---|---|---|---|---|---|---|---|---|
| | Accuracy | Score | Accuracy | Score | Accuracy | Score | Accuracy | Score | Accuracy | Score |
| Full | 65.3 | 3.24 | 48.0 | 2.90 | 30.0 | 1.87 | 12.5 | 1.99 | 43.4 | 2.61 |
| Sim | 59.3 | 2.89 | 47.6 | 2.83 | 23.0 | 1.75 | 15.6 | 2.07 | 40.4 | 2.48 |

Table 19: Quantitative comparison with different judgement prompt. **Red** stands for the best result, **Blue** stands for the second best result.

| Method | Coarse Recognition | | Fine Recognition | | Concept Attribute | | Personalized Caption | | Average | |
|---|---|---|---|---|---|---|---|---|---|---|
| | Accuracy | Score | Accuracy | Score | Accuracy | Score | Accuracy | Score | Accuracy | Score |
| MyVLM | 43.2 | 1.83 | 31.9 | 1.82 | **25.8** | **1.51** | 6.8 | 1.12 | **28.7** | **1.64** |
| Yo'LLaVA | 49.8 | 2.20 | 33.8 | 1.98 | 6.31 | 0.47 | **15.7** | **1.44** | 27.1 | 1.58 |
| MC-LLaVA | **50.7** | **2.23** | **34.5** | **2.01** | 1.80 | 0.18 | **15.2** | 1.42 | 26.7 | 1.55 |
| Ours | **58.3** | **2.56** | **60.0** | **3.07** | **27.3** | **1.63** | 10.3 | **1.63** | **45.2** | **2.40** |

*Proof.* By expanding the definition of the variance $D_b$, we have

$$
\begin{aligned}
D_b &= \frac{1}{H} \sum_{i=1}^{H} (b_i^2 - 2b_i\bar{b} + \bar{b}^2) \\
&= \left( \frac{1}{H} \sum_{i=1}^{H} b_i^2 \right) - \left( \frac{1}{H} \sum_{i=1}^{H} 2b_i\bar{b} \right) + \left( \frac{1}{H} \sum_{i=1}^{H} \bar{b}^2 \right) \\
&= \frac{1}{H} \|b\|^2 - 2\bar{b} \left( \frac{1}{H} \sum_{i=1}^{H} b_i \right) + \frac{1}{H}(H \cdot \bar{b}^2) \\
&= \frac{1}{H} \|b\|^2 - 2\bar{b}^2 + \bar{b}^2 \\
&= \frac{1}{H} \|b\|^2 - \bar{b}^2.
\end{aligned}
\tag{6}
$$

Transform the formula above, and we get

$$
\|b\|^2 = H \cdot (D_b + \bar{b}^2).
\tag{7}
$$

$\square$

**Theorem 2** (Recentering effect of LayerNorm Tuning). *During LayerNorm tuning, we have*

$$
\bar{b} = 0, \|b\|^2 = H \cdot (D_b)
\tag{8}
$$

*Proof.* For a standard LayerNorm, we can formulate it as follow:

$$
y = \frac{x - \mu}{\sigma}, \quad \mu = \frac{1}{H} \sum_{i=1}^{H} x_i, \quad \sigma = \sqrt{\frac{1}{H} \sum_{i=1}^{H} (x_i - \mu)^2},
\tag{9}
$$

where $x = (x_1, \ldots, x_H)^T$ is the input vector of the LayerNorm layer, $y = (y_1, \ldots, y_H)^T$ is the normalized vector. By defining $1_H = (1, \ldots, 1)^T$, the backward propagation with respect to the loss $l$ can be further expressed as:

$$
\begin{aligned}
b = \frac{\partial l}{\partial x} &= \left( \frac{\partial y}{\partial x} + \frac{\partial \mu}{\partial x} \frac{\partial y}{\partial \mu} + \frac{\partial \sigma}{\partial x} \frac{\partial y}{\partial \sigma} \right) \frac{\partial \ell}{\partial y} \\
&= \frac{1}{\sigma} \left( I - \frac{1_H 1_H^T}{H} - \frac{yy^T}{H} \right) \frac{\partial l}{\partial y}
\end{aligned}
\tag{10}
$$

We define $W_1 = \frac{1}{\sigma}(I - \frac{1_H 1_H^T}{H} - \frac{yy^T}{H})$ and let $g = \frac{\partial l}{\partial y} = (g_1, ..., g_H)^T$. Subsequently, we obtain:

$$b = W_1 g,$$

$$\bar{b} = \frac{1}{H}\sum_{i=1}^{H} b_i = \frac{1}{H}(1_H^T b) = \frac{1}{H}(1_H^T W_1)g \tag{11}$$

For $1_H^T W_1$, we have:

$$
\begin{aligned}
1_H^T W_1 &= 1_H^T \left[ \frac{1}{\sigma}\left(I - \frac{1_H 1_H^T}{H} - \frac{yy^T}{H}\right)\right] \\
&= \frac{1}{\sigma}\left[ (1_H^T I) - (1_H^T \frac{1_H 1_H^T}{H}) - (1_H^T \frac{yy^T}{H})\right] \\
&= \frac{1}{\sigma}\left[ 1_H^T - 1_H^T - (\frac{(\sum y_i)y^T}{H})\right] \\
&= 0
\end{aligned}
\tag{12}
$$

Combining Eq. (11) and Eq. (12), we have

$$\bar{b} = \frac{1}{H}(1_H^T b) = 0 \tag{13}$$

Combining Theorem 1 and Eq. (13), we have

$$\|b\|^2 = H \cdot (D_b) \tag{14}$$

$\square$

As demonstrated by Theorem 2, LayerNorm Tuning ensures that the mean of the input gradient remains zero, thereby reducing the overall gradient norm. This allows the gradient direction during optimization to more faithfully reflect the data's directional changes, leading to a smoother optimization trajectory. This finding is consistent with the results in Fig. 12.

**Theorem 3** (Rescaling effect of LayerNorm Tuning). *During LayerNorm tuning, the gradient norm* $\|b\|^2$ *is upper-bounded by* $\frac{HD_g}{\sigma}$:

$$\|b\|^2 \leq \frac{HD_g}{\sigma} \tag{15}$$

*where $D_g = \frac{1}{H}\sum_{i=1}^{H}(g_i - \bar{g})^2$ is the variance of the gradient of the normalized vector y.*

*Proof.* Since $y$ is the normalized vector, we can define a standard orthonormal basis $u_1 = 1_H/\sqrt{H}$, $u_2 = y/\sqrt{H}, \ldots \ldots u_H$ with $1_H$ and $y$. For $W_1$ and $u_i$, we have

$$W_1 u_i = \frac{1}{\sigma}\left(I - \frac{\mathbf{1}_H \mathbf{1}_H^T + yy^T}{H}\right)u_i = \frac{1}{\sigma}\left(u_i - \mathbf{1}_H \frac{\mathbf{1}_H^T u_i}{H} - y\frac{y^T u_i}{H}\right) = \frac{u_i}{\sigma} \tag{16}$$

For $W_1$ and $y$, we have

$$W_1 y = \frac{1}{\sigma}\left(I - \frac{\mathbf{1}_H \mathbf{1}_H^T + yy^T}{H}\right)y = \frac{1}{\sigma}\left(y - \mathbf{1}_H \frac{\mathbf{1}_H^T y}{H} - y\frac{y^T y}{H}\right) = \frac{y - 0 - y}{\sigma} = 0 \tag{17}$$

For $W_1$ and $\mathbf{1}_H$, we have

$$W_1 \mathbf{1}_H = \frac{1}{\sigma}\left(I - \frac{\mathbf{1}_H \mathbf{1}_H^T + yy^T}{H}\right)\mathbf{1}_H = \frac{1}{\sigma}\left(\mathbf{1}_H - \mathbf{1}_H \frac{\mathbf{1}_H^T \mathbf{1}_H}{H} - y\frac{y^T \mathbf{1}_H}{H}\right) = \frac{\mathbf{1}_H - \mathbf{1}_H - 0}{\sigma} = 0 \tag{18}$$

Table 20: Ablation study on caption augmented data.

| Dataset | Coarse Recognition | | Fine Recognition | | Concept Attribute | | Personalized Caption | | Average | |
|---|---|---|---|---|---|---|---|---|---|---|
| | Accuracy | Score | Accuracy | Score | Accuracy | Score | Accuracy | Score | Accuracy | Score |
| Ours w/o caption augmented data | 66.7 | 3.30 | 43.1 | 2.68 | 30.4 | 2.04 | 13.6 | 2.02 | 42.1 | 2.59 |
| Ours | 65.3 | 3.24 | 48.0 | 2.90 | 30.0 | 1.87 | 12.5 | 1.99 | 43.4 | 2.61 |

For $W_i$ and any vector $v = \sum_{i=1}^{H} \lambda_i u$, we have

$$
\begin{aligned}
W_1 v &= \sum_{i=1}^{H} \lambda_i W_1 u_i \\
&= \frac{W_1 \mathbf{1}_H}{\sqrt{H}} + \frac{W_1 y}{\sqrt{H}} + \sum_{i=3}^{H} \lambda_i W_1 u_i \\
&= \sum_{i=3}^{H} \lambda_i \frac{u_i}{\sigma} \\
&= \frac{1}{\sigma} \sum_{i=3}^{H} \lambda_i u_i
\end{aligned}
\tag{19}
$$

Now we can obtain

$$
\|W_1 v\|^2 = \|\frac{1}{\sigma} \sum_{i=3}^{H} \lambda_i u_i\|^2 = \frac{1}{\sigma^2} \sum_{i=3}^{H} \lambda_i^2 \leq \frac{1}{\sigma^2} \sum_{i=1}^{H} \lambda_i^2 = \frac{1}{\sigma^2} \|v\|^2
\tag{20}
$$

According to the definition of the gradient $b$, we have

$$
\begin{aligned}
b &= W_1 g \\
&= W_1 \left[ (g - \overline{g} 1_H) + (\overline{g} 1_H) \right] \\
&= W_1 (g - \overline{g} 1_H) + \overline{g}(W_1 1_H) \\
&= W_1 (g - \overline{g} 1_H)
\end{aligned}
\tag{21}
$$

Combine Eq. (20) and Eq. (21), we have

$$
\begin{aligned}
\|b\|^2 &= \|W_1 (g - \overline{g} 1_H)\|^2 \\
&\leq \frac{1}{\sigma^2} \|(g - \overline{g} 1_H)\|^2 \\
&= \frac{1}{\sigma^2} \sum_{i=1}^{H} (g_i - \overline{g})^2 \\
&= \frac{H D_g}{\sigma}
\end{aligned}
\tag{22}
$$

$\square$

As demonstrated in Theorem 3, LayerNorm Tuning can rescale the gradient norm during each backward propagation. This mechanism is particularly beneficial for VLMs, whose backbones are typically deep LLMs composed of numerous Transformer blocks. For such deep architectures, rescaling effectively prevents the uncontrolled growth of the gradient norm with increasing network depth(Xu et al., 2019), thereby mitigating the risk of gradient explosion.

## K  COMPARISON OF DIFFERENT MODEL SIZE

We report results obtained using LLaVA-1.5-7B as an alternative base model. As shown in Tab. 9, the smaller model generally performs slightly worse than its larger version.

Table 21: Quantitative comparison with Qwen2.5-72B-Instruct Yang et al. (2024). **Red** stands for the best result, **Blue** stands for the second best result.

| Method | Coarse Recognition | | Fine Recognition | | Concept Attribute | | Personalized Caption | | Average | |
|---|---|---|---|---|---|---|---|---|---|---|
| | Accuracy | Score | Accuracy | Score | Accuracy | Score | Accuracy | Score | Accuracy | Score |
| MyVLM | 48.1 | 2.24 | 31.6 | 1.83 | 30.5 | 1.65 | 7.0 | 1.11 | 32.4 | 1.80 |
| Yo'LLaVA | 56.7 | 2.73 | 33.7 | 1.90 | 9.2 | 0.56 | 16.0 | 1.33 | 31.1 | 1.70 |
| MC-LLaVA | 56.5 | 2.72 | 34.7 | 1.98 | 2.5 | 0.21 | 15.9 | 1.52 | 29.7 | 1.67 |
| Ours | 64.9 | 3.06 | 61.2 | 3.07 | 34.4 | 1.83 | 11.8 | 1.45 | 49.2 | 2.55 |

Table 23: Quantitative results of human studies.

| Method | Coarse Recognition | | Fine Recognition | | Concept Attribute | | Personalized Caption | | Average | |
|---|---|---|---|---|---|---|---|---|---|---|
| | Accuracy | Score | Accuracy | Score | Accuracy | Score | Accuracy | Score | Accuracy | Score |
| MyVLM | 47.8 | 2.30 | 27.9 | 1.82 | 29.2 | 1.70 | 10.6 | 1.55 | 31.1 | 1.88 |
| Yo'LLaVA | 56.8 | 2.83 | 29.1 | 1.83 | 9.4 | 0.67 | 18.7 | 1.69 | 29.8 | 1.77 |
| MC-LLaVA | 55.9 | 2.78 | 29.5 | 1.91 | 3.0 | 0.29 | 16.7 | 1.82 | 27.8 | 1.71 |
| Ours | 65.0 | 3.14 | 55.6 | 3.02 | 33.0 | 1.90 | 12.8 | 1.73 | 46.9 | 2.61 |

## L    ABLATION STUDY ON TRAINING DATASET

We validate the effectiveness of our caption augmented training data in Tab. 20. The results demonstrate that our caption augmented training data can enhance the model's Fine Recognition ability.

## M    MORE QUALITATIVE RESULTS

Due to page limitations, the main paper presents only a limited number of qualitative results. To more clearly and comprehensively demonstrate the effectiveness of our method, we provide additional detailed qualitative results in Figs. 17 and 18.

## N    QUANTITATIVE EVALUATION WITH OPEN-SOURCE LLM

We conduct this ablation study for two key reasons: (1) to verify the versatility of our judgment prompt across different judgment LLMs, and (2) our primary judgment LLM (GPT-4o-mini) is not open-source, which poses reproducibility challenges. As a result, we employ the well-known open-source Qwen2.5-72B-Instruct Yang et al. (2024) as an alternative judgment LLM and conduct comprehensive quantitative experiments. As shown in Tab. 21, the results demonstrate similar trends to our original quantitative results under all metrics. This ensures both the accessibility and reproducibility of our evaluation method while confirming the versatility of our judgment prompt.

Table 22: Quantitative results of applying LayerNorm Tuning (LN) to MC-LLaVA An et al. (2024).

| Method | Coarse Recognition | | Fine Recognition | | Concept Attribute | | Personalized Caption | | Average | |
|---|---|---|---|---|---|---|---|---|---|---|
| | Accuracy | Score | Accuracy | Score | Accuracy | Score | Accuracy | Score | Accuracy | Score |
| MC-LLaVA | 55.8 | 2.86 | 24.4 | 1.83 | 4.0 | 0.40 | 16.3 | 2.07 | 26.1 | 1.76 |
| MC-LLaVA + LN | 62.1 | 3.17 | 36.6 | 2.35 | 14.3 | 1.11 | 15.4 | 2.09 | 34.7 | 2.21 |

## O    RESULTS OF HUMAN STUDIES

In order to compare the human evaluation with the LLM evaluation to validate the usage of LLMs as a judge. We conduct human studies with the same instruction as the judgment prompt on our OPBench, and the results are shown in Tab. 23. The findings demonstrate strong alignment between human assessments in the following table and the LLM-as-judge evaluations in Tab. 1.

Table 24: Training time analysis.

| Method | Training Time ↓ |
|--------|-----------------|
| Yo'LLaVA | 11min 40s |
| MC-LLaVA | 11min 45s |
| Ours | **11min 23s** |

## P  VERSATILITY OF THE LAYERNORM TUNING

To further validate the versatility of the LayerNorm tuning strategy, we apply this strategy to MC-LLaVA. As shown in Tab. 22, LayerNorm Tuning demonstrates significant improvements across most evaluation metrics.

## Q  TRAINING EFFICIENCY AND TRAINING LOSS

We conduct a detailed analysis of the training efficiency and training loss for our method and the compared methods. Since MyVLM Alaluf et al. (2024) involves a two-stage training process, we exclude it from this comparison. As shown in Tab. 24, we set the training steps for all methods to 2,000 (personalizing a VLM typically requires 3,000–5,000 steps). We observe that although our method finetunes more parameters than the compared methods, it achieves a reduction in training time. This is because the compared methods adopt soft tokens as extra parameters and incorporate them into the system prompt, which increases computational cost during each forward pass. In contrast, our approach does not incorporate any extra tokens into the system prompt. Although we finetune extra parameters of LayerNorm, the overall training time is slightly reduced.

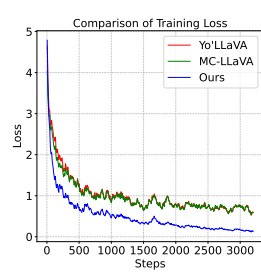

Figure 14: Loss Curve.

Furthermore, we visualize their loss during training. As illustrated in Fig. 14, the compared methods exhibit frequent oscillations at higher loss values. In contrast, our method has a more stable and consistently lower loss curve.

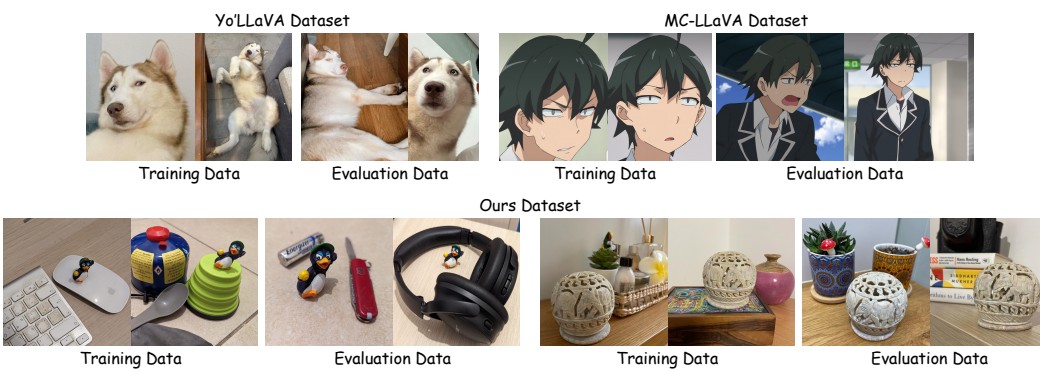

Figure 15: Visualized comparison of different datasets.

## R  MORE DETAILS OF OUR DATASETS

VLM personalization is a relatively new task that lacks a mature and comprehensive benchmark for evaluation. Previous methods, such as Yo'LLaVA Nguyen et al. (2024) and MC-LLaVA An et al. (2024), have proposed their own evaluation datasets. As shown in Fig. 15, these datasets primarily contain images with only the target concept, which brings invisible convenience to both training and evaluation. In contrast, we primarily collect images from Alaluf et al. (2024) to construct our

dataset for two main reasons. (1) We focus on a more challenging real-world setting where each image contains not only the target concept but also identity-irrelevant redundant objects. Images in the dataset of Alaluf et al. (2024) meet our requirements well. (2) This dataset also provides captions for each image, which facilitates our caption augmentation process.

Our dataset includes a total of 335 images for 28 concepts, with 258 used for training. For each concept, which is represented by a set of $N$ training images, we construct our training data as follows:

1. **Negative Recognition Data**: To teach the model to identify the absence of the target concept, we generate negative samples. Following Yo'LLaVA, for each concept, we collect:
   - 100 random images without the target concept.
   - $10 \times N$ images retrieved from the LAION dataset based on CLIP similarity to the target concept, which also serve as hard negative examples.
   - For each of these $100 + 10 \times N$ images, we create one QA pair for negative recognition (e.g., Q: "Is `<sks>` in the image?" A: "No."), totaling $100 + 10 \times N$.
2. **Positive & Contextual Data**: For the $N$ images containing the target concept, we use caption augmentation to generate diverse QA pairs. For each image, we generate 5 QA pairs for each of four categories:
   - Positive Recognition.
   - Localization of the target concept.
   - Description of redundant objects.
   - Interaction between the target concept and redundant objects.
   - This process yields 4 categories $\times$ 5 QA pairs/category = $20 \times N$ training data points from the positive images.

Therefore, for a single concept with $N$ training images, the total number of training data points is $(100 + 10 \times N) + 20 \times N = 100 + 30 \times N$.

## S  LIMITATION AND FUTURE WORKS

Our method effectively learns the visual features of target concepts. However, it struggles to distinguish between objects that are highly similar in appearance, as shown in Fig. 16. Additionally, since the training data does not directly include captions, our method's ability to generate personalized captions remains limited. Furthermore, collecting and organizing real-world image datasets for training and testing is a complex process. Although our dataset incorporates diverse concepts, time constraints prevented us from validating the method on a larger scale.

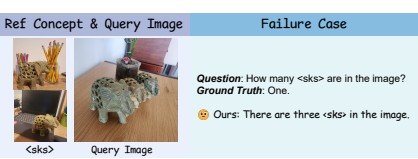

Figure 16: Failure case.

We hope our DVT-LLAVA can inspire future research, especially in enhancing the visual learning ability when personalizing a VLM. Future research directions may focus on: (1) Designing a training-free approach for personalizing large multimodal models (VLMs), (2) Developing evaluation metrics that more accurately and comprehensively assess the performance of personalized VLMs, and (3) Achieving more lightweight and precise personalization of VLMs.

## T  BROADER IMPACT

The purpose of our research is to provide users with a personalized VLM that can perform personalized communication with a specific concept identifier (e.g., `<sks>`). For positive societal impacts, it offers significant societal benefits by providing users with highly personalized suggestions and interactive experiences. For positive societal impacts, as a VLM-based model, it may still potentially generate hallucinations, such as producing incorrect information.

## Coarse Recognition – Positive Recognition

*Question*: Is there a <sks> present in the image?
*Ours*: Yes, there is a<sks> present in the image.

*Question*: Does the image contain a <sks>?
*Ours*: Yes, the image contains a <sks> sitting next to a vase on a shelf.

*Question*: Can you see a <sks> in the picture?
*Ours*: Yes, there is a<sks> visible in the picture next to a large plant ......

## Coarse Recognition – Negative Recognition

*Question*: Is there a <sks> present in the image?
*Ours*: No, there is no <sks> present in the image. Instead, there is a rubber duck figurine inside a blue container or case.

*Question*: Can you see a <sks> in the picture?
*Ours*: No, the image does not contain a <sks>. It features a refrigerator filled with various food items and containers.

*Question*: Does the image contain a <sks>?
*Ours*: No, there is no <sks> in the image. It features a small dog on a leash standing outside near the water.

## Fine Recognition – Counting

*Question*: How many <sks> are in the image?
*Ours*: There is one <sks> in the image.

*Question*: How many <sks> can you find in the image?
*Ours*: There is only one <sks> cup visible in the image.

*Question*: How many <sks> do you see in the picture?
*Ours*: There is only one <sks> visible in the picture.

## Fine Recognition – Redundant Object Recognition

*Question*: Besides the <sks>, what other items can you spot in the image?
*Ours*: In the image, besides the<sks>, I can also see a red Jif peanut butter container on a shelf.

*Question*: Aside from the <sks>, what else is present in the image?
*Ours*: In addition to the<sks>, there are jars of food and candy displayed on a table or countertop in the image.

*Question*: Besides the <sks>, what else is visible in the image?
*Ours*: In addition to the<sks>, there are various other items on the table such as a clock, jewelry, and possibly some collectibles or knick-knacks.

Figure 17: More qualitative results of our methods.

Figure 18: More qualitative results of our methods.

