# OpenReview forum: "DVT-LLaVA: Vision-Language Model Personalization with Disentangled Visual Tuning"
_ICLR.cc/2026/Conference — Submitted to ICLR 2026_

### Official Review · Reviewer_y4iR · 2025-10-31

**Soundness:** 2
**Presentation:** 3
**Contribution:** 2
**Rating:** 4
**Confidence:** 4

**Summary:**

The paper presents DVT-LLaVA, a personalization method for VLMs that learns a disentangled concept representation by jointly training a shared concept-relevant token and per-image concept-irrelevant tokens, while tuning LayerNorm parameters to keep the trainable footprint small. It also injects text-embedding noise to curb shortcut and overfitting. The authors introduce OPBench, an open-ended VQA benchmark for personalized concepts, and report gains over MyVLM, Yo’LLaVA, and MC-LLaVA on recognition and attribute understanding, with reasonable retention of base capabilities. The dataset and setup are sensible and reproducible.

**Strengths:**

- Interesting, low-overhead adaptation: LayerNorm tuning strikes a good accuracy/params balance for personalization.
- Disentanglement idea: Splitting concept-relevant vs. concept-irrelevant tokens encourages visual grounding rather than prompt-only cues.
- Evaluation shift toward realistic use: OPBench’s open-ended judging better reflects real personalization than pure multiple-choice.
- Data & setup: The data composition and training protocol are reasonable and easy to reproduce.

**Weaknesses:**

- Limited comparison breadth: Baselines and benchmarks are too few; stronger/newer methods(RAP-MLLM, SLC, etc.) are under-covered.

- No explicit multi-concept setting: Despite comparing to MC-LLaVA, the paper does not test multi-concept interference.

**Questions:**

- Multi-concept personalization: Please add experiments where multiple learned concepts co-occur (composition/disambiguation), and analyze interference between concept tokens. Any changes needed to the disentanglement design to scale to multiple concepts?

- Benchmark breadth & fairness: Can you evaluate on additional public suites and include more recent, strong baselines—ideally reproduced under your pipeline—or at least standardized, directly comparable metrics?

- Judge reliability: Since the evaluation uses LLM-as-judge, please report judge-model/prompt variance and include a small human study with inter-rater agreement to validate the pipeline.

---

> ### Author Response · Authors · 2025-11-22
> **Response to Reviewer y4iR (1/3)**
>
> We sincerely thank the reviewer for their insightful comments and recognition of our work, particularly for acknowledging the *LayerNorm tuning strategy*, *Disentanglement idea*, and *benchmark* of our method. We have refined the paper, incorporated additional experiments, and clarified the following points in the revised version.
>
> ## **W1 & W2 & Q1 & Q2 More baselines and benchmarks**
>
> **More benchmarks**
>
> - We have included comparison results with additional VLM personalization datasets in **Appendix C**. We include the multi-concept datasets of MC-LLaVA and the dataset of Yo'LLaVA.
>
> **More baselines**
>
> - We include MyVLM, Yo'LLaVA, RAP-MLLM [3], MC-LLaVA, Yo'Chameleon [4] and UnicTokens [5] as the compared baseline methods. Notably, both Yo'Chameleon and UnicTokens are categorized as personalized unified models, which are slightly different from our personalized VLM task. We include these methods for more comprehensive evaluation.
> - We also provide the results of RAP-MLLM on our OPBench. We exclude SCL because it is not fully open source.
>
> **Extend our method to multi-concept scenarios**
>
> - To extend our method to multi-concept scenarios, we first follow MC-LLaVA to assign a specific, concept-relevant token to each personalized concept. We then adapt the data format of the MC-LLaVA dataset to align with our training process. We present all the results in the following four tables.
>
>
> | MC-LLaVA Datset-1 | Choice-V Acc. Single | Choice-V Acc. Multi | Choice-V Acc. Weight | Choice-T Acc. Single | Choice-T Acc. Multi | Choice-T Acc. Weight |
> | :----------------: | :-------------------: | :------------------: | :-------------------: | :-------------------: | :------------------: | :-------------------: |
> | MyVLM             | 0.779                | -                   | 0.779                | -                    | -                   | -                    |
> | Yo’LLaVA          | 0.687                | 0.634               | 0.661                | 0.604                | 0.605               | 0.604                |
> | RAP-MLLM          | 0.832                | 0.690               | 0.784                | **0.709**            | **0.656**           | **0.685**            |
> | MC-LLaVA          | 0.855                | 0.765               | 0.796                | 0.623                | 0.585               | 0.612                |
> | Ours              | **0.857**            | **0.806**           | **0.823**            | 0.546                | 0.515               | 0.536                |
> | Yo'Chameleon      | 0.718                | -                   | 0.718                | 0.561                | -                   | 0.561                |
> | UniCTokens        | 0.753                | -                   | 0.753                | 0.572                | -                   | 0.572                |

---

> > ### Author Response · Authors · 2025-11-22
> > **Response to Reviewer y4iR (2/3)**
> >
> > | MC-LLaVA Datset-2 | VQA BLEU Single | VQA BLEU Multi | VQA BLEU Weight | Captioning Recall Single | Captioning Recall Multi | Captioning Recall Weight |
> > | :----------------: | :--------------: | :-------------: | :--------------: | :-----------------------: | :----------------------: | :-----------------------: |
> > | MyVLM             | 0.640           | -              | **0.640**       | **0.714**                | -                       | 0.714                    |
> > | Yo’LLaVA          | 0.623           | 0.538          | 0.575           | 0.498                    | 0.663                   | 0.548                    |
> > | RAP-MLLM          | 0.424           | 0.423          | 0.424           | 0.711                    | 0.748                   | **0.723**                |
> > | MC-LLaVA          | 0.646           | 0.526          | 0.567           | 0.531                    | 0.708                   | 0.584                    |
> > | Ours              | **0.706**       | **0.603**      | 0.638           | 0.654                    | **0.844**               | 0.711                    |
> > | Yo'Chameleon      | 0.603           | -              | 0.603           | 0.602                    | -                       | 0.602                    |
> > | UniCTokens        | 0.641           | -              | 0.641           | 0.634                    | -                       | 0.634                    |
> >
> > | MC-LLaVA Datset-3 | Rec Single | Rec Multi | Rec Weight |    VG     | Yo'LLaVA Acc. |
> > | :---------------: | :--------: | :-------: | :--------: | :-------: | :-----------: |
> > |       MyVLM       |   0.795    |     -     |   0.795    |   0.688   |     77.92     |
> > |     Yo’LLaVA      | **0.940**  |   0.733   |   0.820    |   0.639   |     92.22     |
> > |     RAP-MLLM      |   0.747    |   0.688   |   0.713    | **0.719** |     94.29     |
> > |     MC-LLaVA      |   0.810    | **0.882** |   0.836    |   0.714   |     95.70     |
> > |       Ours        |   0.927    |   0.779   | **0.873**  |   0.693   |   **96.80**   |
> > |   Yo'Chameleon    |   0.723    |     -     |   0.723    |   0.621   |     74.35     |
> > |    UniCTokens     |   0.742    |     -     |   0.742    |   0.632   |     75.68     |
> >
> > | OPBench  |  CR Acc  | CR Score |  FR Acc  | FR Score |  CA Acc  | CA Score |  PC Acc  | PC Score | Average Accuracy | Average Score |
> > | :------: | :------: | :------: | :------: | :------: | :------: | :------: | :------: | :------: | :--------------: | :-----------: |
> > |  MyVLM   |   48.0   |   2.38   |   22.1   |   1.74   |   26.3   |   1.65   |   12.8   |   1.91   |       28.5       |     1.90      |
> > | Yo'LLaVA |   56.1   |   2.90   |   23.2   |   1.74   |   9.1    |   0.75   | **17.9** |   1.97   |       27.2       |     1.81      |
> > | MC-LLaVA |   55.8   |   2.86   |   24.4   |   1.83   |   4.0    |   0.40   |   16.3   | **2.07** |       26.1       |     1.76      |
> > | RAP-MLLM |   48.3   |   2.33   |   4.02   |   0.49   | **50.7** | **2.55** |   2.03   |   0.18   |       27.2       |     1.56      |
> > |   Ours   | **65.3** | **3.24** | **48.0** | **2.90** |   30.0   |   1.87   |   12.5   |   1.99   |     **43.4**     |   **2.61**    |
> >
> > For MC-LLaVA dataset, although not explicitly designed for multi-concept scenarios, our method demonstrates strong performance across various visual evaluation tasks, including Choice-V, VQA, and Rec. Our method also achieves superior performance in the Yo'LLaVA dataset. These results indicate our method's capability to navigate more complex, multi-concept settings.
> >
> > [3] RAP: Retrieval-Augmented Personalization for Multimodal Large Language Models. In CVPR, 2025.
> >
> > [4] Yo'Chameleon: Personalized Vision and Language Generation. In CVPR, 2025.
> >
> > [5] UniCTokens: Boosting Personalized Understanding and Generation via Unified Concept Tokens. In NIPS, 2025.

---

> > > ### Author Response · Authors · 2025-11-22
> > > **Response to Reviewer y4iR (3/3)**
> > >
> > > ## **Q3 Judge reliability**
> > >
> > > For real-world open-ended question answering, LLM-as-a-judge is a widely adopted evaluation protocol [2]. We further provide three groups of experiments to avoid evaluation bias.
> > >
> > > **Evaluation results with different LLMs**
> > >
> > > To mitigate potential bias from relying on a single large language model (LLM), we adopt Qwen2.5-72B-Instruct for evaluation in **Appendix H**. We present the results in the table below, where CR stands for `Coarse Recognition`, FR stands for `Fine Recognition`, CA stands for `Concept Attribute`, PC stands for `Personalized Caption`.
> > >
> > > | Method   | CR Acc   | CR Score | FR Acc   | FR Score | CA Acc   | CA Score | PC Acc   | PC Score | Average Accuracy | Average Score |
> > > | :-------: | :-------: | :-------: | :-------: | :-------: | :-------: | :-------: | :-------: | :-------: | :---------------: | :------------: |
> > > | MyVLM    | 48.1     | 2.24     | 31.6     | 1.83     | 30.5     | 1.65     | 7.0      | 1.11     | 32.4             | 1.80          |
> > > | Yo'LLaVA | 56.7     | 2.73     | 33.7     | 1.90     | 9.2      | 0.56     | **16.0** | 1.33     | 31.1             | 1.70          |
> > > | MC-LLaVA | 56.5     | 2.72     | 34.7     | 1.98     | 2.5      | 0.21     | 15.9     | **1.52** | 29.7             | 1.67          |
> > > | Ours     | **64.9** | **3.06** | **61.2** | **3.07** | **34.4** | **1.83** | 11.8     | 1.45     | **49.2**         | **2.55**      |
> > >
> > > The results demonstrate a similar trend to our original quantitative results in Tab. 1.
> > >
> > > **Evaluation results with a different judgment prompt**
> > >
> > > To mitigate potential bias from the judgment prompt, we use a different judgment prompt for evaluation in **Appendix I**. We present the results in the table below. The results also show a similar trend to the table above and the original results in Tab. 1.
> > >
> > > | Method   | CR Acc   | CR Score | FR Acc   | FR Score | CA Acc   | CA Score | PC Acc   | PC Score | Average Accuracy | Average Score |
> > > | :-------: | :-------: | :-------: | :-------: | :-------: | :-------: | :-------: | :-------: | :-------: | :---------------: | :------------: |
> > > | MyVLM    | 43.2     | 1.83     | 31.9     | 1.82     | 25.8     | 1.51     | 6.8      | 1.12     | **28.7**         | **1.64**      |
> > > | Yo'LLaVA | 49.8     | 2.20     | 33.8     | 1.98     | 6.31     | 0.47     | **15.7** | 1.44     | 27.1             | 1.58          |
> > > | MC-LLaVA | **50.7** | **2.23** | **34.5** | **2.01** | 1.80     | 0.18     | 15.2     | 1.42     | 26.7             | 1.55          |
> > > | Ours     | **58.3** | **2.56** | **60.0** | **3.07** | **27.3** | **1.63** | 10.3     | **1.63** | **45.2**         | **2.40**      |
> > >
> > > **Evaluation results of human study**
> > >
> > > To compare the human evaluation with the LLM evaluation to validate the usage of LLMs-as-a-judge. We have included human studies results in **Appendix O**. To ensure metric consistency, we provide the human experts with the judgment prompt. The results are presented below, which demonstrates strong alignment between human and LLM evaluation.
> > >
> > > | Method | CR Acc | CR Score | FR Acc | FR Score | CA Acc | CA Score | PC Acc | PC Score | Average Accuracy | Average Score |
> > > | :---: | :---: | :---: | :---: | :---: | :---: | :---: | :---: | :---: | :---: | :---: |
> > > | MyVLM | 47.8 | 2.30 | 27.9 | 1.82 | 29.2 | 1.70 | 10.6 | 1.55 | 31.1 | 1.88 |
> > > | Yo'LLaVA | 56.8 | 2.83 | 29.1 | 1.83 | 9.4 | 0.67 | **18.7** | 1.69 | 29.8 | 1.77 |
> > > | MC-LLaVA | 55.9 | 2.78 | 29.5 | 1.91 | 3.0 | 0.29 | 16.7 | **1.82** | 27.8 | 1.71 |
> > > | Ours | **65.0** | **3.14** | **55.6** | **3.02** | **33.0** | **1.90** | 12.8 | 1.73 | **46.9** | **2.61** |
> > >
> > > [2] From Generation to Judgment: Opportunities and Challenges of LLM-as-a-judge. In EMNLP, 2025.

---

### Official Review · Reviewer_nkQx · 2025-11-01

**Soundness:** 3
**Presentation:** 3
**Contribution:** 2
**Rating:** 4
**Confidence:** 4

**Summary:**

This paper conducts research on three core pain points in the personalization of vision-language models (VLMs): existing methods rely on the shortcut of "memorizing textual descriptions" rather than learning visual features, concept-relevant/irrelevant information is easily entangled in complex backgrounds, and multiple-choice evaluation benchmarks cannot accurately measure performance in open-set scenarios. To this end, the authors propose the DVT-LLaVA framework, whose core designs include three parts: 1) Disentangled Visual Representation Learning: by jointly training shared concept-relevant tokens  and image-specific concept-irrelevant tokens , visual features are separated without textual descriptions of target concepts; 2) LayerNorm Tuning Strategy: only fine-tuning the LayerNorm layers in VLMs (accounting for approximately 0.003% of parameters) to enhance the ability to learn complex concepts while avoiding catastrophic forgetting; 3) Text Embedding Augmentation: adding controllable uniform noise to query embeddings to alleviate overfitting. In addition, the authors construct the OPBench open-style benchmark, covering 28 concepts, 335 images, and 3115 open VQA questions, evaluating from four dimensions: coarse recognition, fine recognition, concept attributes, and personalized captioning. Experiments show that DVT-LLaVA achieves an average accuracy of 43.4% on OPBench, significantly outperforming baseline methods such as MyVLM (28.5%) and Yo’LLaVA (27.2%), while maintaining the performance of the pre-trained model on benchmarks like POPE and RealWorldQA, verifying the balance between personalized learning and pre-trained knowledge retention.

**Strengths:**

1. Accurate and quantitative problem positioning: Instead of only qualitatively criticizing the "text memorization shortcut", it clarifies the performance gap of existing methods in visual tasks (e.g., counting) and textual tasks (e.g., color description) through controlled experiments with/without textual descriptions (Tab.5), providing quantitative basis for subsequent method design.

2. Excellent balance between parameter efficiency and performance: LayerNorm Tuning involves only 0.003% of model parameters (Fig.10), with training time (11min23s) slightly lower than Yo’LLaVA (11min40s). Without textual descriptions, the fine recognition accuracy (48.0%) is more than twice that of Yo’LLaVA (23.2%) (Tab.1), balancing practical deployment and performance advantages.

3. Benchmark design close to real needs: OPBench adopts an open VQA format to avoid the "random guessing" flaw of multiple-choice benchmarks. Its questions cover real user interaction scenarios such as "redundant object description" and "concept interaction" (e.g., "What other items are there besides  in Fig.7"), making it more practically valuable than existing benchmarks.

**Weaknesses:**

1. Lack of interpretability and robustness in the disentanglement mechanism: The separation of  and  is only qualitatively proven through attention visualization (Fig.8), without using quantitative tools such as Concept Activation Vectors (CAV) or feature mutual information to verify the disentanglement effect. Additionally, it does not test "concept neighbor interference" scenarios (e.g.,  = "red pig piggy bank",  = "red cup"), failing to prove that  will not mistakenly absorb key visual features of  or that the two interact, leaving the core innovation without essential support.

2. Weak generalization and theoretical basis of LayerNorm Tuning: It does not explain the theoretical mechanism by which "LayerNorm Tuning promotes visual learning" (e.g., how mean/variance adjustment affects vision-language feature alignment), only verifying it retroactively through performance improvement, resulting in insufficient theoretical support.

3. Lack of adversarial and practical tests in the evaluation system: It does not design confusing concept pairs (e.g., "apple" vs. "tomato"), which are precisely the core pain points of VLM personalization. Meanwhile, it does not evaluate the model’s performance in multi-concept learning.

**Questions:**

1. The number of benchmarks used in the paper is insufficient, and the results are not enough to illustrate the effectiveness of the proposed scheme. Additional comparisons with other methods on common benchmarks are needed. (E.g., Yo'Chameleon[1], UniCTokens[2] and MyVLM/ Yo'LlaVA / MC-LLaVA Datasets, if you do these experiments, I will consider raise my score.)

2. In the "concept neighbor interference" scenario ( = "red pig piggy bank",  = "red cup"), what is the feature purity of  (e.g., accuracy of probing and classifying target regions)?

3. On VLMs with different architectures such as Qwen-VL or GPT-4V, what is the performance degradation of LayerNorm Tuning? Is it necessary to adjust the noise coefficient α or the selection of tuned layers?

4. In tests with confusing concept pairs ("apple" vs. "tomato"), how much will DVT-LLaVA’s coarse recognition accuracy decrease? This can also reflect whether  truly learns comprehensive features.

[1] YoChameleon: Personalized Vision and Language Generation, https://arxiv.org/abs/2504.20998

[2] UniCTokens: Boosting Personalized Understanding and Generation via Unified Concept Tokens, https://arxiv.org/abs/2505.14671

**Details Of Ethics Concerns:**

The research content of the paper focuses on VLM personalization methods and evaluation benchmarks. The training data is derived from a public academic dataset (Alaluf et al. 2024), involving no private images, human subjects, or harmful application scenarios; the use of LLMs (e.g., GPT-4o-mini as an evaluation judge) is clearly disclosed, with no issues related to data compliance, research integrity, or academic misconduct.

---

> ### Author Response · Authors · 2025-11-22
> **Response to Reviewer nkQx (1/4)**
>
> We sincerely thank the reviewer for their insightful comments and recognition of our work, particularly for acknowledging the *motivation*, *efficiency*, and *benchmark* of our method. We have refined the paper, incorporated additional experiments, and clarified the following points in the revised version.
>
> ## **W1 Quantitative analysis of disentanglement**
>
> Following your suggestion, we quantitatively evaluate the disentanglement between concept-relevant and concept-irrelevant tokens using three metrics: `Cosine Similarity` (CS), `Feature Mutual Information` (FMI), and `Distance Correlation` (DC). A detailed analysis is provided in **Appendix E**. The quantitative results are presented as follows:
>
> | Metric | CS (Mean) | CS (Std) | FMI (Mean) | FMI (Std) | DC (Mean) | DC (Std) |
> | :-----: | :-------: | :------: | :--------: | :-------: | :-------: | :------: |
> | Value  |   0.033   |  0.033   |   0.007    |   0.001   |   0.050   |  0.021   |
>
> The results indicate a low similarity and minimal statistical correlation between concept-relevant and concept-irrelevant tokens. This demonstrates that our disentangled visual representation learning approach has effectively disentangled these two components.
>
> ## **W1 & W3 & Q2 & Q4 Concept neighbor interference scenario**
>
> ### **Evaluation under concept neighbor interference scenarios**
>
> - We agree that learning confusing concept pairs is the core pain point of VLM personalization. Our OPBench is able to address such a challenging and real-world setting. Among our benchmark, some redundant objects are very similar to the target concepts in color or shape. For example, the "red piggy bank" and "red potted plant" in **Fig. 13 of Appendix H**. We term these objects "concept neighbors" like the reviewer.
> - To evaluate our model's performance in the presence of such concept neighbors, we carefully select a subset called OPBench-Sim from our OPBench. OPBench-Sim contains concepts with concept neighbors in the evaluation image. With this benchmark, we can gain a deeper understanding of how our method performs when dealing with concept neighbors.
> - We have added the related analysis to **Appendix H** and present the quantitative results in the table below, where CR stands for `Coarse Recognition`, FR stands for `Fine Recognition`, CA stands for `Concept Attribute`, PC stands for `Personalized Caption`.
>
> | Method | OPBench | CR Acc | CR Score | FR Acc | FR Score | CA Acc | CA Score | PC Acc | PC Score | Average Accuracy | Average Score |
> | :----: | :-----: | :----: | :------: | :----: | :------: | :----: | :------: | :----: | :------: | :--------------: | :-----------: |
> |  Ours  |  Full   |  65.3  |   3.24   |  48.0  |   2.90   |  30.0  |   1.87   |  12.5  |   1.99   |       43.4       |     2.61      |
> |  Ours  |   Sim   |  59.3  |   2.89   |  47.6  |   2.83   |  23.0  |   1.75   |  15.6  |   2.07   |       40.4       |     2.48      |
> |        |         |        |          |        |          |        |          |        |          |                  |               |
>
> As we mentioned in the **Limitation part of Appendix S**, distinguishing between highly similar objects is a very difficult task even for humans. Under this circumstance, our method can still accurately learn the visual features of the target concept without significant performance degradation.
>
> ### **Feature purity with the concept neighbor interference**
>
> We provide visualized results of the "red piggy bank" and "red potted plant" in **Fig. 13** of **Appendix H**. The results indicate that our method robustly probes the target concept within the image across different scenarios, despite interference from concept neighbors. This outcome highlights our core innovation: learning consistent visual features across diverse scenarios rather than relying on the shortcut of memorizing textual descriptions.
>
> ### **How much does coarse recognition decrease**
>
> As shown in the table above, the Coarse Recognition accuracy of our method decreases from *65.3% to 59.3%*, while the corresponding score falls from *3.24 to 2.89*. This performance degradation is considered acceptable, given that concept neighbor interference scenarios represent a particularly challenging setting.

---

> > ### Author Response · Authors · 2025-11-22
> > **Response to Reviewer nkQx (2/4)**
> >
> > ## **W2 Theoretical analysis of LayerNorm tuning**
> >
> > - Thank you for your illuminating suggestions. We provide deeper insights into our LayerNorm tuning strategy by analyzing gradients during training, as the gradient is a crucial indicator of the quality and stability of the training process.
> > - We first visualize the trend of the gradient norm under various training strategies (e.g., Self-attn, MLP, LoRa and our LayerNorm tuning) in **Fig. 12**. The results show that the gradient norm of our LayerNorm tuning is smaller and more stable compared to other training strategies.
> > - In order to understand the reason behind this situation, we provide a theoretical analysis of the *recentering* and *rescaling* effect of LayerNorm tuning.
> > - The recentering effect ensures the mean of the input gradient remains zero, thereby reducing the overall gradient norm. Concurrently, the rescaling effect imposes a clear upper bound on the gradient norm for the LayerNorm layer. Together, these two effects facilitate a more constrained and smoother optimization trajectory during LayerNorm tuning. This not only helps to mitigate overfitting but also reduces the risk of gradient explosion during the training process.
> > - We have added the detailed proof in **Appendix J**, where the *recentering effect* is proven in Theorem 2 and the *rescaling effect* is proven in Theorem 2.
> >
> >
> > ## **W3 & Q1 More related methods and benchmarks**
> >
> >
> > - We have included comparison results with additional VLM personalization datasets in **Appendix C**. We include the multi-concept datasets of MC-LLaVA and the dataset of Yo'LLaVA. Due to a partial overlap between our OPBench and the MyVLM datasets, the latter was excluded. We prioritize our OPBench because it comprises a more extensive collection of the real-world, open-ended evaluation samples that are central to our analysis.
> > - We include MyVLM, Yo'LLaVA, RAP-MLLM [2], MC-LLaVA, Yo'Chameleon [3] and UnicTokens [4] as the compared baseline methods. Notably, both Yo'Chameleon and UnicTokens are categorized as personalized unified models, which are slightly different from our personalized VLM task. We include these methods for more comprehensive evaluation. We present the results in the following three tables.
> > - For MC-LLaVA dataset, although not explicitly designed for multi-concept scenarios, our method demonstrates strong performance across various visual evaluation tasks, including Choice-V, VQA, and Rec. Our method also achieves superior performance in the Yo'LLaVA dataset. These results indicate our method's capability to navigate more complex, multi-concept settings.

---

> > > ### Author Response · Authors · 2025-11-22
> > > **Response to Reviewer nkQx (3/4)**
> > >
> > > | MC-LLaVA Datset-1 | Choice-V Acc. Single | Choice-V Acc. Multi | Choice-V Acc. Weight | Choice-T Acc. Single | Choice-T Acc. Multi | Choice-T Acc. Weight |
> > > | :----------------: | :-------------------: | :------------------: | :-------------------: | :-------------------: | :------------------: | :-------------------: |
> > > | MyVLM             | 0.779                | -                   | 0.779                | -                    | -                   | -                    |
> > > | Yo’LLaVA          | 0.687                | 0.634               | 0.661                | 0.604                | 0.605               | 0.604                |
> > > | RAP-MLLM          | 0.832                | 0.690               | 0.784                | **0.709**            | **0.656**           | **0.685**            |
> > > | MC-LLaVA          | 0.855                | 0.765               | 0.796                | 0.623                | 0.585               | 0.612                |
> > > | Ours              | **0.857**            | **0.806**           | **0.823**            | 0.546                | 0.515               | 0.536                |
> > > | Yo'Chameleon      | 0.718                | -                   | 0.718                | 0.561                | -                   | 0.561                |
> > > | UniCTokens        | 0.753                | -                   | 0.753                | 0.572                | -                   | 0.572                |
> > >
> > > | MC-LLaVA Datset-2 | VQA BLEU Single | VQA BLEU Multi | VQA BLEU Weight | Captioning Recall Single | Captioning Recall Multi | Captioning Recall Weight |
> > > | :----------------: | :--------------: | :-------------: | :--------------: | :-----------------------: | :----------------------: | :-----------------------: |
> > > | MyVLM             | 0.640           | -              | **0.640**       | **0.714**                | -                       | 0.714                    |
> > > | Yo’LLaVA          | 0.623           | 0.538          | 0.575           | 0.498                    | 0.663                   | 0.548                    |
> > > | RAP-MLLM          | 0.424           | 0.423          | 0.424           | 0.711                    | 0.748                   | **0.723**                |
> > > | MC-LLaVA          | 0.646           | 0.526          | 0.567           | 0.531                    | 0.708                   | 0.584                    |
> > > | Ours              | **0.706**       | **0.603**      | 0.638           | 0.654                    | **0.844**               | 0.711                    |
> > > | Yo'Chameleon      | 0.603           | -              | 0.603           | 0.602                    | -                       | 0.602                    |
> > > | UniCTokens        | 0.641           | -              | 0.641           | 0.634                    | -                       | 0.634                    |
> > >
> > >
> > > | MC-LLaVA Datset-3 | Rec Single | Rec Multi | Rec Weight | VG        | Yo'LLaVA Acc. |
> > > | :----------------: | :---------: | :--------: | :---------: | :--------: | :------------: |
> > > | MyVLM             | 0.795      | -         | 0.795      | 0.688     | 77.92         |
> > > | Yo’LLaVA          | **0.940**  | 0.733     | 0.820      | 0.639     | 92.22         |
> > > | RAP-MLLM          | 0.747      | 0.688     | 0.713      | **0.719** | 94.29         |
> > > | MC-LLaVA          | 0.810      | **0.882** | 0.836      | 0.714     | 95.70         |
> > > | Ours              | 0.927      | 0.779     | **0.873**  | 0.693     | **96.80**     |
> > > | Yo'Chameleon      | 0.723      | -         | 0.723      | 0.621     | 74.35         |
> > > | UniCTokens        | 0.742      | -         | 0.742      | 0.632     | 75.68         |
> > >
> > > [3] RAP: Retrieval-Augmented Personalization for Multimodal Large Language Models. In CVPR, 2025.
> > >
> > > [4] Yo'Chameleon: Personalized Vision and Language Generation. In CVPR, 2025.
> > >
> > > [5] UniCTokens: Boosting Personalized Understanding and Generation via Unified Concept Tokens. In NIPS, 2025.

---

> > > > ### Author Response · Authors · 2025-11-22
> > > > **Response to Reviewer nkQx (4/4)**
> > > >
> > > > ## **Q3 Ablation study of different architectures, noise weight factor and LayerNorm tuning**
> > > >
> > > > Thanks for your reasonable suggestion. We select two representative architectures as our backbones: Phi3 [6] and Qwen [7]. We conduct a detailed ablation study regarding the noise weight factor $\alpha$ of our text embedding augmentation and our LayerNorm tuning. We have added a detailed analysis to **Appendix G**.
> > > >
> > > > ### **Noise weight factor**
> > > >
> > > > We first perform an ablation study on the noise weight factor $\alpha$ of our text embedding augmentation. The results are presented below.
> > > >
> > > >
> > > > | Phi3 $\alpha$ |  CR Acc  | CR Score |  FR Acc  | FR Score |  CA Acc  | CA Score |  PC Acc  | PC Score | Average Accuracy | Average Score |
> > > > | :-----: | :---: | :---: | :---: | :---: | :---: | :---: | :---: | :---: | :---: | :---: |
> > > > |       0       |   64.5   |   3.04   |   45.4   |   2.42   |   21.6   |   1.30   |   5.23   |   0.58   |       39.2       |     2.07      |
> > > > |       5       |   65.0   |   3.10   |   49.5   |   2.56   |   20.8   |   1.19   |   4.34   |   0.63   |       40.5       |     2.11      |
> > > > |      15       |   67.4   |   3.22   |   54.0   |   2.75   |   22.4   |   1.31   |   2.32   |   0.56   |       43.0       |     2.23      |
> > > > |      25       | **69.1** | **3.31** | **54.7** | **2.76** | **29.8** | **1.61** | **8.21** | **1.05** |     **46.2**     |   **2.39**    |
> > > > |      35       |   68.3   |   3.24   |   52.5   |   2.65   |   18.7   |   1.14   |   8.03   |   1.08   |       42.4       |     2.22      |
> > > >
> > > > | Qwen $\alpha$ |  CR Acc  | CR Score |  FR Acc  | FR Score |  CA Acc  | CA Score |  PC Acc  | PC Score | Average Accuracy | Average Score |
> > > > | :----: | :---: | :---: | :---: | :---: | :---: | :---: | :---: | :---: | :---: | :---: |
> > > > |       0       |   78.1   |   3.83   |   51.8   |   2.62   |   27.3   |   1.50   | **18.6** | **1.54** |       48.1       |     2.50      |
> > > > |       5       |   75.7   |   3.71   |   52.3   |   2.65   |   31.1   |   1.63   |   15.8   |   1.51   |       48.3       |     2.52      |
> > > > |      15       |   79.8   |   3.77   |   63.3   |   3.03   |   37.4   |   1.97   |   14.5   |   1.50   |       54.9       |     2.76      |
> > > > |      25       | **88.3** | **4.11** | **63.4** | **3.12** |   37.7   | **2.08** |   12.2   |   1.33   |     **56.8**     |   **2.88**    |
> > > > |      35       |   86.8   |   3.95   |   56.1   |   2.79   | **39.2** |   2.00   |   9.82   |   1.32   |       53.7       |     2.70      |
> > > >
> > > > - The results show that incorporating text embedding augmentation always yields superior performance compared to models without text embedding augmentation.
> > > > - Furthermore, both architectures achieved optimal performance at $\alpha = 25$, which is consistent with the configuration used in our main experiments. This finding further substantiates the robustness and parameter-insensitivity of our proposed text embedding augmentation strategy.
> > > >
> > > > ### **LayerNorm tuning**
> > > >
> > > > For LayerNorm tuning, we evaluate three distinct configurations to assess the impact of finetuning LayerNorm layers:
> > > > 1. No, a baseline in which the LayerNorm layers are not finetuned.
> > > > 2. Half, where only half of the LayerNorm layers are finetuned.
> > > > 3. Full, where all LayerNorm layers are finetuned.
> > > > We present the results in the tables below.
> > > >
> > > > | Phi3 |  CR Acc  | CR Score |  FR Acc  | FR Score |  CA Acc  | CA Score |  PC Acc  | PC Score | Average Accuracy | Average Score |
> > > > | :---: | :------: | :------: | :------: | :------: | :------: | :------: | :------: | :------: | :--------------: | :-----------: |
> > > > |  No  |   43.4   |   2.14   |   14.2   |   0.98   |   4.88   |   0.32   |   5.23   |   0.87   |       18.0       |     1.09      |
> > > > | Half |   59.1   |   2.85   |   46.0   |   2.51   |   14.3   |   1.01   |   7.66   |   0.97   |       36.5       |     2.02      |
> > > > | Full | **69.1** | **3.31** | **54.7** | **2.76** | **29.8** | **1.61** | **8.21** | **1.05** |     **46.2**     |   **2.39**    |
> > > >
> > > > | Qwen |  CR Acc  | CR Score |  FR Acc  | FR Score |  CA Acc  | CA Score |  PC Acc  | PC Score | Average Accuracy | Average Score |
> > > > | :---: | :------: | :------: | :------: | :------: | :------: | :------: | :------: | :------: | :--------------: | :-----------: |
> > > > |  No  |   57.1   |   2.80   |   25.3   |   1.53   |   14.0   |   0.84   |   10.0   |   1.25   |       28.5       |     1.64      |
> > > > | Half |   79.3   |   3.72   |   49.7   |   2.75   |   28.1   |   1.47   |   17.6   |   1.49   |       47.7       |     2.51      |
> > > > | Full | **88.3** | **4.11** | **63.4** | **3.12** | **37.7** | **2.08** | **12.2** | **1.33** |     **56.8**     |   **2.88**    |
> > > >
> > > > The results demonstrate that finetuning LayerNorm layers consistently yields superior performance compared to the baseline. Furthermore, the Full configuration, which involved finetuning all LayerNorm layers, achieved the optimal results.
> > > >
> > > > [6] Phi-3 Technical Report: A Highly Capable Language Model Locally on Your Phone. In Arxiv, 2024.
> > > >
> > > > [7] Qwen technical report. In Arxiv, 2023.

---

### Official Review · Reviewer_FMgH · 2025-11-01

**Soundness:** 3
**Presentation:** 4
**Contribution:** 3
**Rating:** 6
**Confidence:** 3

**Summary:**

This paper focuses on VLM personalization and proposes a method to enhance the visual learning capability for target concepts in complex background. Specifically, the authors introduce concept-relevant and concept-irrelevant tokens to learn disentangled visual representations for target concepts, enabling finer control over concept-specific features. To mitigate overfitting, they tune the LayerNorm layers and employ a text embedding augmentation strategy. Additionally, they introduce OPBench, a benchmark for evaluation in an open-set setting, which includes an open-style VQA format.

**Strengths:**

1. The core idea is straightforward and the method is easy-to-follow.
2. The experiments demonstrate notable performance improvements, supported by good visualization results that validate the paper’s claims.

**Weaknesses:**

1. The study would benefit from comparisons with additional VLM personalization datasets, such as those from Yo’LLaVA and MyVLM.
2. The text embedding augmentation and concept-irrelevant tokens lead to significant accuracy drops in Personalized Caption (Table 2). Could you explain it?

**Questions:**

1. What methods ensure disentangled visual representations for target concepts in VLMs? Are there losses or regularizers for concept-relevant vs. irrelevant tokens?

2. During inference, what are the VLM's inputs? Are all concept-irrelevant tokens included?

3. Does the number of irrelevant tokens affect performance? Since this likely varies with background complexity, are there strategies to handle such variability?

---

> ### Author Response · Authors · 2025-11-22
> **Response to Reviewer FMgH (1/4)**
>
> We sincerely thank the reviewer for their insightful comments and recognition of our work, particularly for acknowledging the *method*, *experiments*, and *visualization results* of our method. We have refined the paper, incorporated additional experiments, and clarified the following points in the revised version.
>
> ## **W1 Comparisons with additional datasets**
>
> - We have included comparison results with additional VLM personalization datasets in **Appendix C**. We include the datasets of MC-LLaVA and Yo'LLaVA. Due to a partial overlap between our OPBench and the MyVLM datasets, the latter was excluded. We prioritize our OPBench because it comprises a more extensive collection of the real-world, open-ended evaluation samples that are central to our analysis.
> - We include MyVLM, Yo'LLaVA, RAP-MLLM [3], MC-LLaVA, Yo'Chameleon [4] and UnicTokens [5] as the compared baseline methods. Notably, both Yo'Chameleon and UnicTokens are categorized as personalized unified models, which are slightly different from our personalized VLM task. We include these methods for more comprehensive evaluation. We present the results in the following three tables.
> - For MC-LLaVA dataset, although not explicitly designed for multi-concept scenarios, our method demonstrates strong performance across various visual evaluation tasks, including Choice-V, VQA, and Rec. Our method also achieves superior performance in the Yo'LLaVA dataset. These results indicate our method's capability to navigate more complex, multi-concept settings.

---

> > ### Author Response · Authors · 2025-11-22
> > **Response to Reviewer FMgH (2/4)**
> >
> > | MC-LLaVA Datset-1 | Choice-V Acc. Single | Choice-V Acc. Multi | Choice-V Acc. Weight | Choice-T Acc. Single | Choice-T Acc. Multi | Choice-T Acc. Weight |
> > | :----------------: | :-------------------: | :------------------: | :-------------------: | :-------------------: | :------------------: | :-------------------: |
> > | MyVLM             | 0.779                | -                   | 0.779                | -                    | -                   | -                    |
> > | Yo’LLaVA          | 0.687                | 0.634               | 0.661                | 0.604                | 0.605               | 0.604                |
> > | RAP-MLLM          | 0.832                | 0.690               | 0.784                | **0.709**            | **0.656**           | **0.685**            |
> > | MC-LLaVA          | 0.855                | 0.765               | 0.796                | 0.623                | 0.585               | 0.612                |
> > | Ours              | **0.857**            | **0.806**           | **0.823**            | 0.546                | 0.515               | 0.536                |
> > | Yo'Chameleon      | 0.718                | -                   | 0.718                | 0.561                | -                   | 0.561                |
> > | UniCTokens        | 0.753                | -                   | 0.753                | 0.572                | -                   | 0.572                |
> >
> > | MC-LLaVA Datset-2 | VQA BLEU Single | VQA BLEU Multi | VQA BLEU Weight | Captioning Recall Single | Captioning Recall Multi | Captioning Recall Weight |
> > | :----------------: | :--------------: | :-------------: | :--------------: | :-----------------------: | :----------------------: | :-----------------------: |
> > | MyVLM             | 0.640           | -              | **0.640**       | **0.714**                | -                       | 0.714                    |
> > | Yo’LLaVA          | 0.623           | 0.538          | 0.575           | 0.498                    | 0.663                   | 0.548                    |
> > | RAP-MLLM          | 0.424           | 0.423          | 0.424           | 0.711                    | 0.748                   | **0.723**                |
> > | MC-LLaVA          | 0.646           | 0.526          | 0.567           | 0.531                    | 0.708                   | 0.584                    |
> > | Ours              | **0.706**       | **0.603**      | 0.638           | 0.654                    | **0.844**               | 0.711                    |
> > | Yo'Chameleon      | 0.603           | -              | 0.603           | 0.602                    | -                       | 0.602                    |
> > | UniCTokens        | 0.641           | -              | 0.641           | 0.634                    | -                       | 0.634                    |
> >
> > | MC-LLaVA Datset-3 | Rec Single | Rec Multi | Rec Weight | VG        | Yo'LLaVA Acc. |
> > | :----------------: | :---------: | :--------: | :---------: | :--------: | :------------: |
> > | MyVLM             | 0.795      | -         | 0.795      | 0.688     | 77.92         |
> > | Yo’LLaVA          | **0.940**  | 0.733     | 0.820      | 0.639     | 92.22         |
> > | RAP-MLLM          | 0.747      | 0.688     | 0.713      | **0.719** | 94.29         |
> > | MC-LLaVA          | 0.810      | **0.882** | 0.836      | 0.714     | 95.70         |
> > | Ours              | 0.927      | 0.779     | **0.873**  | 0.693     | **96.80**     |
> > | Yo'Chameleon      | 0.723      | -         | 0.723      | 0.621     | 74.35         |
> > | UniCTokens        | 0.742      | -         | 0.742      | 0.632     | 75.68         |
> >
> > [3] RAP: Retrieval-Augmented Personalization for Multimodal Large Language Models. In CVPR, 2025.
> >
> > [4] Yo'Chameleon: Personalized Vision and Language Generation. In CVPR, 2025.
> >
> > [5] UniCTokens: Boosting Personalized Understanding and Generation via Unified Concept Tokens. In NIPS, 2025.

---

> > > ### Author Response · Authors · 2025-11-22
> > > **Response to Reviewer FMgH (3/4)**
> > >
> > > ## **W2 Accuracy drops in personalized caption**
> > >
> > > - The primary objective of text embedding augmentation and concept-irrelevant tokens is to enhance our model's visual understanding of the target concept. Both strategies successfully improve the model's performance on Coarse Recognition, Fine Recognition, and Concept Attribute.
> > > - However, image captioning presents a more significant challenge, as it requires the model to accurately describe the broader context surrounding the target concept, rather than just identifying the concept itself. These two strategies make the model focus more on the visual information of the target concept. Consequently, they inadvertently compromise the model's ability to generate contextually rich and descriptive captions.
> > > - Moreover, as discussed in the **Limitation part of Appendix S**, our training data does not directly include caption generation QA pairs, which limits the model's ability to produce personalized captions.
> > >
> > > ## **Q1 Disentanglement of concept relevant and irrelevant tokens**
> > >
> > > - In our initial design, we did not apply explicit losses or regularizers to concept-relevant and concept-irrelevant tokens. The concept-relevant token is shared across all the training images, while the concept-irrelevant token is specific to each training image. With our caption augmentation data, we enable the concept-relevant token to learn a consistent visual feature of the target concept across all training images, and the concept-irrelevant token to learn the background features specific to each image. As a result, the concept-relevant and concept-irrelevant tokens are implicitly disentangled within our framework. **Fig. 8** in the main paper also visually illustrates this disentanglement.
> > > - Inspired by your suggestion, we add a cosine similarity loss to explicitly disentangle the concept-relevant tokens and concept-irrelevant tokens through minimizing the similarity between these two types of tokens. We present the results in the table below, where CR stands for `Coarse Recognition`, FR stands for `Fine Recognition`, CA stands for `Concept Attribute`, PC stands for `Personalized Caption`. The results show that our method does not benefit from the explicit disentanglement.
> > >
> > > |   Method    | CR Acc | CR Score | FR Acc | FR Score | CA Acc | CA Score | PC Acc | PC Score | Average Accuracy | Average Score |
> > > | :---------: | :----: | :------: | :----: | :------: | :----: | :------: | :----: | :------: | :--------------: | :-----------: |
> > > |    Ours     |  69.1  |   3.31   |  54.7  |   2.76   |  29.8  |   1.61   |  8.21  |   1.05   |       46.3       |     2.40      |
> > > | Ours w/ cos |  72.9  |   3.48   |  52.4  |   2.71   | 24.76  |   1.47   |  11.4  |   1.22   |       45.5       |     2.40      |
> > >
> > > - To figure out the deeper reason behind this phenomenon, we further conduct quantitative analysis of the disentanglement. We adopt `Cosine Similarity` (CS), `Feature Mutual Information` (FMI), `Distance Correlation` (DC) to measure the disentanglement effect between concept-relevant tokens and concept-irrelevant tokens. We have included the detailed analysis in **Appendix E**. The quantitative results are as follows:
> > >
> > > | Metric | CS (Mean) | CS (Std) | FMI (Mean) | FMI (Std) | DC (Mean) | DC (Std) |
> > > | :----: | :-------: | :------: | :--------: | :-------: | :-------: | :------: |
> > > | Value  |   0.033   |  0.033   |   0.007    |   0.001   |   0.050   |  0.021   |
> > >
> > > The results indicate a low similarity and minimal statistical correlation between concept-relevant and concept-irrelevant tokens. This may explain the ineffectiveness of explicit disentanglement, as our method already achieves effective disentanglement between concept-relevant and concept-irrelevant tokens.
> > >
> > > ## **Q2 Input during inference**
> > >
> > > To maintain vocabulary consistency between training and inference for reproducibility, we include all concept-irrelevant tokens during the inference stage in the current experimental setting. However, the logits of all concept-irrelevant tokens are set to negative infinity to avoid their effect during inference. For practical usage, we can adopt only concept-relevant tokens during inference.

---

> > > > ### Author Response · Authors · 2025-11-22
> > > > **Response to Reviewer FMgH (4/4)**
> > > >
> > > > ## **Q3 Ablation study on the number of concept-irrelevant tokens**
> > > >
> > > > - Thanks for your suggestion. During training, too many concept-irrelevant tokens would cause the model to focus too much on learning redundant objects in the background, while impairing the learning of the visual features of the target concept.
> > > > - Since we are not concerned with the specific details of various complex background elements, we employ a single concept-irrelevant token to represent them collectively.
> > > > - We conduct an ablation study on the number of concept irrelevant tokens $M$ with our OPBench, and the results are shown in the following table. The results show that more concept irrelevant tokens do not bring obvious performance improvement to our method.
> > > >
> > > > |       | Average Accuracy | Average Score |
> > > > | :---: | :--------------: | :-----------: |
> > > > | $M=1$ |       64.1       |     3.55      |
> > > > | $M=2$ |       61.1       |     3.39      |
> > > > | $M=3$ |       58.3       |     3.21      |
> > > > | $M=4$ |       59.8       |     3.20      |

---

### Official Review · Reviewer_P1qH · 2025-11-01

**Soundness:** 3
**Presentation:** 3
**Contribution:** 3
**Rating:** 4
**Confidence:** 4

**Summary:**

This paper studies the problem of personalizing vision-language models (VLMs) to make the model better recognize and describe that concept in visual-language tasks. Accordingly, this work introduces DVT-LLaVA with disentangled visual tuning, layernorm tuning, and text embedding augmentation. Experiments show the effectiveness of the proposed method.

**Strengths:**

* This work identifies a realistic and understudied failure mode in personalized VLMs, which over relay on textual memory.

* The proposed method requires simple yet effective tuning, which could be a low-cost solution compared with existing fine-tuning methods.

* This work introduce a benchmark, which should be supported if the benchmark can be properly released to the academy.

**Weaknesses:**

* It seems that there lacks sufficient evidence for the disentanglement. Quantitative results, rather than the qualitative analysis are expected.

* Although the method avoids explicit textual descriptions of the personalized concept, the caption-derived QA pairs still carry strong linguistic priors (object names, contexts, co-occurrences). Thus, improvements might still stem from language-side memorization rather than purely visual disentanglement.

* OPBench includes only 335 images / 28 concepts / 3115 QAs, which is relatively small, and relies on LLM-as-a-judge (GPT-4o-mini) for scoring. Such evaluation may introduce style or prompt bias.

**Questions:**

* Is that possible to analyze the effect of decoding parameters (temperature, beam size, length penalty) to confirm that the gain is not merely due to text generation bias.

* It seems that the results show that LayerNorm tuning performs better and avoids overfitting compared to LoRA or MLP tuning, but the reason behind this is unclear.

---

> ### Author Response · Authors · 2025-11-22
> **Response to Reviewer P1qH (1/4)**
>
> We sincerely thank the reviewer for their insightful comments and recognition of our work, particularly for acknowledging the *motivation*, *effectiveness*, and *benchmark* of our method. We have refined the paper, incorporated additional experiments, and clarified the following points in the revised version.
> ## **W1 Quantitative analysis of disentanglement**
>
> Thanks for your suggestion, We adopt `Cosine Similarity` (CS), `Feature Mutual Information` (FMI), and `Distance Correlation` (DC [1]) to quantitatively measure the disentanglement effect between concept-relevant tokens and concept-irrelevant tokens. We have included the detailed analysis in **Appendix E**. The quantitative results are as follows:
>
> | Metric | CS (Mean) | CS (Std) | FMI (Mean) | FMI (Std) | DC (Mean) | DC (Std) |
> | :----: | :-------: | :------: | :--------: | :-------: | :-------: | :------: |
> | Value  |   0.033   |  0.033   |   0.007    |   0.001   |   0.050   |  0.021   |
>
> The results indicate a low similarity and minimal statistical correlation between concept-relevant and concept-irrelevant tokens. This demonstrates that our disentangled visual representation learning approach has effectively disentangled these two components.
>
> [1] Measuring and testing dependence by correlation of distances. In The Annals of Statistics, 2007.
>
> ## **W2 Clarification on the linguistic priors of our caption generated data**
>
> - For object names, all QA pairs in our dataset strictly avoid object names with rich linguistic priors (e.g., red piggy bank). These object names are substituted with a newly introduced language token (e.g., `<sks>`).
> - For contexts and concurrences, all text descriptions only refer to objects outside the target concept, such as other redundant objects that appear together, and the environment in which the target concept is located. However, remembering this kind of textual information can not benefit the model's visual understanding of the target concept. Taking **Fig. 5** in the main paper as an example, even if the model remembers the context of the target concept (e.g., next to a bottle, on a red tablecloth), such textual information provides minimal help for the model's visual understanding of the target concept in a new environment.
>
> As a result, our method prevents the model from direct access to the linguistic information of the target concept. In this case, the model has to learn the visual feature of the target concept. We also include the ablation study of the decoding parameters in **Q1** below to demonstrate the robustness of our learned visual features.

---

> > ### Author Response · Authors · 2025-11-22
> > **Response to Reviewer P1qH (2/4)**
> >
> > ## **W3 Benchmark scale and Bias of LLM-as-a-judge**
> >
> > ### **Benchmark scale**
> > - The design of OPBench is motivated by two primary objectives:
> > 	1. We aim to prevent models from achieving high accuracy by merely exploiting cues in multiple-choice options, rather than demonstrating a comprehension of the target concept.
> > 	2. We would like to facilitate the evaluation of models in real-world, open-ended VQA scenarios.
> > - In order to achieve the desired effect, we need to design a large number of questions that accurately reflect the model's visual understanding of the target concept. Simultaneously, we need to manually annotate some of these questions to ensure the quality of the answers. As we mentioned in the **Limitation part of Appendix S**, creating such a high-quality dataset is more resource-intensive than a multiple-choice benchmark. These factors limit the size of our dataset.
> > - We have also included quantitative comparison results on MC-LLaVA and Yo'LLaVA datasets in **Appendix C** for more comprehensive evaluation. We present the results in the table below. Our method demonstrates strong performance on these datasets.
> > - We appreciate the suggestion and plan to expand OPBench to multi-concept scenarios in the future. This extension is straightforward within our current pipeline. We will also release our OPBench to benefit the community.
> >
> > | MC-LLaVA Datset-1 | Choice-V Acc. Single | Choice-V Acc. Multi | Choice-V Acc. Weight | Choice-T Acc. Single | Choice-T Acc. Multi | Choice-T Acc. Weight |
> > | :----------------: | :-------------------: | :------------------: | :-------------------: | :-------------------: | :------------------: | :-------------------: |
> > | MyVLM             | 0.779                | -                   | 0.779                | -                    | -                   | -                    |
> > | Yo’LLaVA          | 0.687                | 0.634               | 0.661                | 0.604                | 0.605               | 0.604                |
> > | RAP-MLLM          | 0.832                | 0.690               | 0.784                | **0.709**            | **0.656**           | **0.685**            |
> > | MC-LLaVA          | 0.855                | 0.765               | 0.796                | 0.623                | 0.585               | 0.612                |
> > | Ours              | **0.857**            | **0.806**           | **0.823**            | 0.546                | 0.515               | 0.536                |
> > | Yo'Chameleon      | 0.718                | -                   | 0.718                | 0.561                | -                   | 0.561                |
> > | UniCTokens        | 0.753                | -                   | 0.753                | 0.572                | -                   | 0.572                |
> >
> > | MC-LLaVA Datset-2 | VQA BLEU Single | VQA BLEU Multi | VQA BLEU Weight | Captioning Recall Single | Captioning Recall Multi | Captioning Recall Weight |
> > | :----------------: | :--------------: | :-------------: | :--------------: | :-----------------------: | :----------------------: | :-----------------------: |
> > | MyVLM             | 0.640           | -              | **0.640**       | **0.714**                | -                       | 0.714                    |
> > | Yo’LLaVA          | 0.623           | 0.538          | 0.575           | 0.498                    | 0.663                   | 0.548                    |
> > | RAP-MLLM          | 0.424           | 0.423          | 0.424           | 0.711                    | 0.748                   | **0.723**                |
> > | MC-LLaVA          | 0.646           | 0.526          | 0.567           | 0.531                    | 0.708                   | 0.584                    |
> > | Ours              | **0.706**       | **0.603**      | 0.638           | 0.654                    | **0.844**               | 0.711                    |
> > | Yo'Chameleon      | 0.603           | -              | 0.603           | 0.602                    | -                       | 0.602                    |
> > | UniCTokens        | 0.641           | -              | 0.641           | 0.634                    | -                       | 0.634                    |
> >
> > | MC-LLaVA Datset-3 | Rec Single | Rec Multi | Rec Weight | VG        | Yo'LLaVA Acc. |
> > | :----------------: | :---------: | :--------: | :---------: | :--------: | :------------: |
> > | MyVLM             | 0.795      | -         | 0.795      | 0.688     | 77.92         |
> > | Yo’LLaVA          | **0.940**  | 0.733     | 0.820      | 0.639     | 92.22         |
> > | RAP-MLLM          | 0.747      | 0.688     | 0.713      | **0.719** | 94.29         |
> > | MC-LLaVA          | 0.810      | **0.882** | 0.836      | 0.714     | 95.70         |
> > | Ours              | 0.927      | 0.779     | **0.873**  | 0.693     | **96.80**     |
> > | Yo'Chameleon      | 0.723      | -         | 0.723      | 0.621     | 74.35         |
> > | UniCTokens        | 0.742      | -         | 0.742      | 0.632     | 75.68         |

---

> > > ### Author Response · Authors · 2025-11-22
> > > **Response to Reviewer P1qH (3/4)**
> > >
> > > ### **Bias of LLM-as-a-judge**
> > >
> > > For real-world open-ended question answering, LLM-as-a-judge is a widely adopted evaluation protocol [2]. We further provide three groups of experiments to avoid evaluation bias.
> > >
> > > **Evaluation results with different LLMs**
> > >
> > > To mitigate potential bias from relying on a single large language model (LLM), we adopt Qwen2.5-72B-Instruct for evaluation in **Appendix N**. We present the results in the table below, where CR stands for `Coarse Recognition`, FR stands for `Fine Recognition`, CA stands for `Concept Attribute`, PC stands for `Personalized Caption`.
> > >
> > > | Method   | CR Acc   | CR Score | FR Acc   | FR Score | CA Acc   | CA Score | PC Acc   | PC Score | Average Accuracy | Average Score |
> > > | :-------: | :-------: | :-------: | :-------: | :-------: | :-------: | :-------: | :-------: | :-------: | :---------------: | :------------: |
> > > | MyVLM    | 48.1     | 2.24     | 31.6     | 1.83     | 30.5     | 1.65     | 7.0      | 1.11     | 32.4             | 1.80          |
> > > | Yo'LLaVA | 56.7     | 2.73     | 33.7     | 1.90     | 9.2      | 0.56     | **16.0** | 1.33     | 31.1             | 1.70          |
> > > | MC-LLaVA | 56.5     | 2.72     | 34.7     | 1.98     | 2.5      | 0.21     | 15.9     | **1.52** | 29.7             | 1.67          |
> > > | Ours     | **64.9** | **3.06** | **61.2** | **3.07** | **34.4** | **1.83** | 11.8     | 1.45     | **49.2**         | **2.55**      |
> > >
> > > The results demonstrate similar a trend to our original quantitative results in Tab. 1.
> > >
> > >
> > >
> > > **Evaluation results with a different judgment prompt**
> > >
> > > To mitigate potential bias from the judgment prompt, we use a different judgment prompt for evaluation in **Appendix I**. We present the results in the table below. The results also show a similar trend to the table above and the original results in Tab. 1.
> > >
> > > | Method   | CR Acc   | CR Score | FR Acc   | FR Score | CA Acc   | CA Score | PC Acc   | PC Score | Average Accuracy | Average Score |
> > > | :-------: | :-------: | :-------: | :-------: | :-------: | :-------: | :-------: | :-------: | :-------: | :---------------: | :------------: |
> > > | MyVLM    | 43.2     | 1.83     | 31.9     | 1.82     | 25.8     | 1.51     | 6.8      | 1.12     | **28.7**         | **1.64**      |
> > > | Yo'LLaVA | 49.8     | 2.20     | 33.8     | 1.98     | 6.31     | 0.47     | **15.7** | 1.44     | 27.1             | 1.58          |
> > > | MC-LLaVA | **50.7** | **2.23** | **34.5** | **2.01** | 1.80     | 0.18     | 15.2     | 1.42     | 26.7             | 1.55          |
> > > | Ours     | **58.3** | **2.56** | **60.0** | **3.07** | **27.3** | **1.63** | 10.3     | **1.63** | **45.2**         | **2.40**      |
> > >
> > > **Evaluation results of human study**
> > >
> > > To compare the human evaluation with the LLM evaluation to validate the usage of LLMs-as-a-judge. We have included human studies results in **Appendix O**. To ensure metric consistency, we provide the human experts with the judgment prompt. The results are presented below, which demonstrate strong alignment between human and LLM evaluation.
> > >
> > > | Method | CR Acc | CR Score | FR Acc | FR Score | CA Acc | CA Score | PC Acc | PC Score | Average Accuracy | Average Score |
> > > | :---: | :---: | :---: | :---: | :---: | :---: | :---: | :---: | :---: | :---: | :---: |
> > > | MyVLM | 47.8 | 2.30 | 27.9 | 1.82 | 29.2 | 1.70 | 10.6 | 1.55 | 31.1 | 1.88 |
> > > | Yo'LLaVA | 56.8 | 2.83 | 29.1 | 1.83 | 9.4 | 0.67 | **18.7** | 1.69 | 29.8 | 1.77 |
> > > | MC-LLaVA | 55.9 | 2.78 | 29.5 | 1.91 | 3.0 | 0.29 | 16.7 | **1.82** | 27.8 | 1.71 |
> > > | Ours | **65.0** | **3.14** | **55.6** | **3.02** | **33.0** | **1.90** | 12.8 | 1.73 | **46.9** | **2.61** |
> > >
> > > [2] From Generation to Judgment: Opportunities and Challenges of LLM-as-a-judge. In EMNLP, 2025.

---

> > > > ### Author Response · Authors · 2025-11-22
> > > > **Response to Reviewer P1qH (4/4)**
> > > >
> > > > ## **Q1 Ablation study of decoding parameters**
> > > >
> > > > - During inference, we follow previous methods to employ deterministic sampling, which can ensure the reproducibility of the results.
> > > > - To analyze the impact of text generation bias on the model's outputs, we conduct an ablation study on three key decoding parameters: *length penalty*, *number of beams*, and *temperature*.
> > > > - We have added the detailed analysis in **Appendix F** and provide the results in the table below.
> > > >
> > > > | Length Penalty | CR Acc | CR Score | FR Acc | FR Score | CA Acc | CA Score | PC Acc | PC Score | Average Accuracy | Average Score |
> > > > | :-------------: | :-----: | :-------: | :-----: | :-------: | :-----: | :-------: | :-----: | :-------: | :---------------: | :------------: |
> > > > | 0.8            | 73.3   | 3.53     | 53.7   | 2.75     | 28.6   | 1.60     | 9.16   | 0.99     | 46.8             | 2.44          |
> > > > | 1.0            | 72.8   | 3.50     | 53.6   | 2.73     | 29.4   | 1.66     | 7.67   | 0.92     | 46.6             | 2.43          |
> > > > | 1.2            | 73.0   | 3.51     | 53.8   | 2.75     | 28.7   | 1.62     | 5.17   | 0.84     | 46.2             | 2.43          |
> > > >
> > > > | Numbers of Beams | CR Acc | CR Score | FR Acc | FR Score | CA Acc | CA Score | PC Acc | PC Score | Average Accuracy | Average Score |
> > > > | :---------------: | :----: | :------: | :----: | :------: | :----: | :------: | :----: | :------: | :--------------: | :-----------: |
> > > > | 1                | 69.1   | 3.31     | 54.7   | 2.76     | 29.8   | 1.61     | 8.21   | 1.05     | 46.2             | 2.39          |
> > > > | 3                | 72.7   | 3.50     | 53.5   | 2.75     | 28.6   | 1.63     | 7.26   | 0.92     | 46.3             | 2.43          |
> > > > | 5                | 72.6   | 3.50     | 53.6   | 2.72     | 27.4   | 1.58     | 6.72   | 0.87     | 45.9             | 2.40          |
> > > >
> > > > | Temperature | CR Acc | CR Score | FR Acc | FR Score | CA Acc | CA Score | PC Acc | PC Score | Average Acc | Average Score |
> > > > | :----------: | :----: | :------: | :----: | :------: | :----: | :------: | :----: | :------: | :---------: | :-----------: |
> > > > | 0.2         |  70.4  |   3.38   |  51.2  |   2.65   |  26.7  |   1.48   |  8.63  |   0.89   |    44.6     |     2.32      |
> > > > | 0.5         |  70.6  |   3.40   |  50.6  |   2.62   |  23.3  |   1.36   |  7.02  |   0.84   |    43.3     |     2.28      |
> > > > | 0.8         |  70.7  |   3.37   |  47.0  |   2.48   |  22.7  |   1.32   |  8.15  |   0.88   |    42.0     |     2.21      |
> > > >
> > > > The results indicate that our model performs consistently across different decoding parameters, suggesting that it has robustly learned the visual features of the target concept and is not significantly influenced by text generation bias.
> > > >
> > > > ## **Q2 Reasons of the better performance of LayerNorm tuning**
> > > >
> > > > - Thank you for your illuminating suggestions. We provide deeper insights into our LayerNorm tuning strategy by analyzing gradients during training, as the gradient is a crucial indicator of the quality and stability of the training process.
> > > > - We first visualize the trend of the gradient norm under various training strategies (e.g., Self-attn, MLP, LoRa and our LayerNorm tuning) in **Fig. 12**. The results show that the gradient norm of our LayerNorm tuning is smaller and more stable compared to other training strategies.
> > > > - In order to understand the reason behind this situation, we provide a theoretical analysis of the *recentering* and *rescaling* effect of LayerNorm tuning.
> > > > - The recentering effect ensures the mean of the input gradient remains zero, thereby reducing the overall gradient norm. Concurrently, the rescaling effect imposes a clear upper bound on the gradient norm for the LayerNorm layer. Together, these two effects facilitate a more constrained and smoother optimization trajectory during LayerNorm tuning. This not only helps to mitigate overfitting but also reduces the risk of gradient explosion during the training process.
> > > > - We have added the detailed proof in **Appendix J**, where the *recentering effect* is proven in Theorem 2 and the *rescaling effect* is proven in Theorem 2.

---

### Author Response · Authors · 2025-11-22
**General response**

We extend our sincere gratitude to all the reviewers (**R1-P1qH**, **R2-FMgH**, **R3-nkQx** and **R4-y4iR**) for their insightful and considerate reviews, which help us to emphasize the contributions of our approach.

We are very encouraged to hear that the reviewers recognized our **clear motivation** (R1, R2, R3), **effectiveness** (R1, R2, R4), **efficiency** (R1, R3, R4), **experiments** (R2), the **advantageous results** we presented (R2) and the **benchmark** we proposed (R1, R3, R4).

We would also like to express our sincere gratitude to the reviewers for their insightful identification of areas where our manuscript could be strengthened. We have taken all the suggestions carefully and updated our previous version. In the revised manuscript, we have made the following **changes**:

1. Added quantitative analysis of disentanglement in **Appendix E**. (R1, R2, R3)
2. Added more quantitative comparison results on other related benchmark datasets in **Appendix C**. (R1, R2, R3, R4)
3. Refined evaluation results with different LLMs in **Appendix N**. (R1, R4)
4. Added evaluation results with different judgment prompts in **Appendix I**. (R1, R4)
5. Refined evaluation results of human study in **Appendix O**. (R1, R4)
6. Added ablation study of decoding parameters in **Appendix F**. (R1)
7. Added theoretical analysis of LayerNorm tuning in **Appendix J**. (R1, R3)
8. Added analysis under similar concepts in **Appendix H**. (R3)
9. Added ablation study of different architectures, noise weight factor, and LayerNorm tuning in **Appendix G**. (R3)

We sincerely hope to **engage in further discussion with the reviewers to ensure all concerns have been fully addressed**. If any aspects of our work remain unclear, we welcome **any further feedback** to help improve our manuscript. Thank you very much again!

---

### Meta-Review · Area_Chair_82bE · 2026-01-07

**Summary:**

This paper introduces DVT-LLaVA, a framework that enhances personalized vision-language models by learning disentangled visual representations to prevent shortcut learning from textual data, while also establishing a new benchmark for accurately evaluating open-set performance.

While AC appreciates the authors' diligent response, the core concerns persist. The work presents a technically sound but incremental contribution. The reviews consistently placed it at the margin of acceptance, citing limitations in novelty and evaluation breadth. The rebuttal, though thorough, does not alter this fundamental assessment. Given ICLR's competitive threshold, AC recommends rejection.

**Reviewer Concerns:**

--outstanding--: It seems that there lacks sufficient evidence for the disentanglement. Quantitative results, rather than the qualitative analysis are expected.

**Reviewer Scores:**

Regarding the comment that the study would benefit from comparisons with additional VLM personalization datasets, such as those from Yo’LLaVA and MyVLM, the authors explicitly provided additional experimental results during the rebuttal. These results directly address the reviewer’s concern. Had the reviewer been able to participate fully in the discussion, the AC believes this clarification would likely have led to a revision of their initial score.

---

### Decision · Program_Chairs · 2026-01-26

Reject